# The matrisome landscape controlling in vivo germ cell fates

Aqilah Amran[1,2,3,4,5], Lara Pigatto[1,2,3,4,5], Johanna Farley [1,2,3], Rasoul Godini[4], Roger Pocock [4] ✉ & Sandeep Gopal [1,2,3,4] ✉

The developmental fate of cells is regulated by intrinsic factors and the extracellular environment. The extracellular matrix (matrisome) delivers chemical and mechanical cues that can modify cellular development. However, comprehensive understanding of how matrisome factors control cells in vivo is lacking. Here we show that specific matrisome factors act individually and collectively to control germ cell development. Surveying development of undifferentiated germline stem cells through to mature oocytes in the *Caenorhabditis elegans* germ line enabled holistic functional analysis of 443 conserved matrisome-coding genes. Using high-content imaging, 3D reconstruction, and cell behavior analysis, we identify 321 matrisome genes that impact germ cell development, the majority of which (>80%) are undescribed. Our analysis identifies key matrisome networks acting autonomously and non-autonomously to coordinate germ cell behavior. Further, our results demonstrate that germ cell development requires continual remodeling of the matrisome landscape. Together, this study provides a comprehensive platform for deciphering how extracellular signaling controls cellular development and anticipate this will establish new opportunities for manipulating cell fates.

The organic extracellular matrix, or matrisome, is comprised of structural proteins, enzymes, growth factors, saccharides, and other small molecules[1]. The matrisome regulatory output is determined by the independent and collective functions of matrisome components, many of which are post-translationally modified[2]. Deciphering how the matrisome controls cell fate and behavior requires systematic dissection of matrisome components in a tractable in vivo model. Here, we exploit the *Caenorhabditis elegans* germ line to investigate how the matrisome controls cell fate and behavior in a complex tissue.

Primordial germ cells are the precursors of mammalian gametes[3]. The roles of certain matrisome molecules in primordial germ cell development in mammals have previously been described[4,5]. Yet, how the matrisome collectively controls germ cell fate and behavior in vivo is unclear. In *C. elegans*, individual matrisome molecules such as

growth factors, structural proteins and proteoglycans can function in germ cell regulation[6–8]. However, systematic analysis of matrisome functions in the development of germline stem cells into gametes is lacking. The adult *C. elegans* germ line initiates with a self-renewing population of mitotic germline stem cells within the distal progenitor zone (PZ). As the cells move proximally, they enter prophase of meiosis I at the transition zone (TZ), before differentiating into oocytes and sperm[9–12]. Development of germline stem cells into gametes is controlled by both intrinsic and extracellular signaling[9,13–15]. While the role of intrinsic signaling has been extensively studied, the role of matrisome signaling remains unclear[9]. The ability of adult *C. elegans* hermaphrodites to produce oocytes from undifferentiated germline stem cells provides an avenue to explore matrisome functions at all stages of germ cell development in vivo[9].

[1]Department of Experimental Medical Science, Lund University, Lund, Sweden. [2]Lund Stem Cell Center, Lund University, Lund, Sweden. [3]Lund Cancer Center, Lund University, Lund, Sweden. [4]Development and Stem Cells Program, Monash Biomedicine Discovery Institute. Department of Anatomy and Developmental Biology, Monash University, Melbourne, VIC, Australia. [5]These authors contributed equally: Aqilah Amran, Lara Pigatto. ✉ e-mail: roger.pocock@monash.edu; sandeep.gopal@med.lu.se

The structural integrity of the *C. elegans* germ line is maintained by five pairs of sheath cells and basement membrane[11]. While the *C. elegans* basement membrane matrix is not fully characterized, fluorescent reporters for matrisome structural proteins (collagens, laminins, nidogen, fibulin) and receptors (proteoglycans) are expressed within it[16]. However, the expression of other matrisome proteins such as catabolic enzymes, proteases and secreted ligands in the basement membrane is not well characterized. These molecules are crucial as they control basement membrane dynamics[16,17], and their significance in germline development is often evidenced by the sterility or reduced brood size[18–20]. Previous studies showed that loss of basement membrane structural components and certain metalloproteinases can cause impaired germline development and distal tip cell migration[21–26]. Yet, how the matrisome post-developmentally impacts germ cells is poorly understood.

In silico characterization of matrisome proteins reveals significant conservation between *C. elegans* and other organisms[27]. Here, we used RNA interference (RNAi) to systematically profile the function of 443 genes from the *C. elegans* matrisome with orthologs in humans (see methods)[27]. Our high-throughput analysis of >3500 germ lines and >7 million cells surveyed cell behavior from immature undifferentiated germline stem cells through to mature gametes. We identified matrisome gene functions by examining germ cell number, nuclear and plasma membrane morphology, protein distribution, cytoskeletal structure, apoptosis, and oocyte morphology. Combining experimental and bioinformatic analysis, we identified critical matrisome molecules and their collective, specific roles during the formation of gametes from undifferentiated germ cells.

## Results

### Profiling matrisome-associated functions in the *C. elegans* germ line

The *C. elegans* matrisome is highly conserved, with ~60% of proteins having predicted human orthologs (Supplementary Fig. 1 and Supplementary Data 1)[27]. Gene ontology (GO) analysis shows that *C. elegans* matrisome components have similar biological functions to mammals (Supplementary Fig. 1 and Supplementary Data 1). We categorized *C. elegans* matrisome genes into families according to established curated databases (Supplementary Fig. 1 and Supplementary Data 1). Overrepresentation analysis of comparable mammalian orthologs suggests that *C. elegans* matrisome genes function in 250 signaling pathways, with 13 pathways having a high confidence level (Supplementary Fig. 1 and Supplementary Data 1).

We generated a plasmid library of the 443 matrisome genes conserved in humans (see methods for gene selection) to enable RNAi gene silencing by feeding (Fig. 1 and Supplementary Data 1)[28,29]. We systematically silenced individual matrisome genes using short-term RNAi for 16 h from the late larval stage 4 (L4) hermaphrodite stage (Fig. 1a). This timing was selected to preclude potential early developmental defects caused by matrisome gene RNAi knockdown and long-term secondary impacts. Only a limited number of matrisome genes have previously described germline defects, most of which were caused by either early RNAi (from larval stage L1) or gene mutations (Supplementary Data 2). However, supporting the important roles for matrisome genes in the germ line is the sterility and reduced brood size observed following their loss (Supplementary Data 2). We used immunofluorescence, 3D reconstruction confocal microscopy, and high-throughput cellular phenotypic analysis to examine germline structure, germ cell number and germ cell cycle, nuclear and protein distribution, plasma membrane structure, apoptosis, and oocyte morphology in 1-day old adult hermaphrodites following RNAi (Fig. 1a). This expansive phenotyping identified 321 matrisome-coding genes that are important for germ cell development (Fig. 1b–d). We first investigated eight phenotypes as numbered—i) PZ cell number, ii) TZ cell number iii) multinucleated cells (MNC) in the pachytene region, iv)

apoptosis, v) oocyte nuclear morphology, vi) oocyte blebbing, vii) multinucleated oocytes and viii) gonad morphology (Supplementary Fig. 2). The number of distinct phenotypes observed per gene knockdown ranged from one to four (Fig. 1b). To confirm the phenotypes observed, selected genes were studied using genetic mutants and secondary RNAi analysis (Supplementary Fig. 3). We observed localized germ line defects (214 genes) limited to the distal, pachytene or oocyte regions, and more global defects (107 genes) identified in multiple germline regions or in gonad structure (Fig. 1c, d). A large number of matrisome gene knockdowns caused defects in the distal to pachytene region, suggesting that these cells are sensitive to matrisome perturbation (Defects i–iv Fig. 1e). Further, we identified 100 genes important for oocyte development (Defects v–vii Fig. 1e). A small number of gene knockdowns resulted in defective gonad placement, where gonads failed to maintain their normal U-shaped morphology (Defect viii in Fig. 1e). This defect resembles gonad defects caused by impaired distal cell migration during development[30]. However, such migration defects are an unlikely cause of the gonad morphology phenotype as RNAi was applied at late L4 when the gonad development nears completion.

Matrisome proteins can act locally or coordinate long-range communication between tissues. Using a published spatial gene expression dataset, we investigated whether matrisome genes expressed in the gonad control germline development (Fig. 1f)[31]. We identified limited correlation between matrisome genes that are expressed at the same location as the phenotype observed following silencing in the germ line (Fig. 1f and Supplementary Data 3). However, we identified 121 genes that are expressed in the gonad and cause a germline phenotype (Supplementary Data 3). Only a small percentage of genes showed a phenotype at the locus of the expression (Fig. 1f). Further, we identified 200 matrisome genes that are not expressed in the gonad yet cause a germline phenotype following RNAi (Supplementary Data 3). This suggests a major role for non-cell-autonomous control of germline biology by the matrisome (Fig. 1f and Supplementary Data 3), though the precise origins of this control are unclear. In addition, RNAi can have systemic impact leading to physiological changes in the animal that affect the germline. Next, we investigated how many of the matrisome genes that we found control the germ line have not been previously linked with germ cell/gamete development. We compared our list of matrisome gene hits to genes associated with GO terms "gamete generation" (GO:0007276) and its child term "germ cell development" (GO:0007281) in *C. elegans*, *Drosophila*, zebrafish, mice and humans (Supplementary Data 4). This identified 15 matrisome genes and/or their orthologs previously associated with germ cell development and gamete generation, leaving 306 (out of 321 genes with phenotype) genes not previously associated with these GO terms (Supplementary Data 4).

### Matrisome components control distal germ cell behavior

*C. elegans* germ cell production begins at the distal PZ, followed by the TZ where germ cells are in early meiosis[32,33]. These regions contain defined numbers of germ cells in wild-type animals[33,34]. To study the impact of matrisome genes on germ cell behavior, we quantified germ cell number in the PZ and TZ following RNAi (Supplementary Data 5). We focus here on 218 matrisome genes excluding the C-type LECtins (CLECs), a large group of predicted matrisome proteins (Supplementary Data 1), which will be addressed separately in the manuscript. We found that RNAi silencing of 37 genes affected PZ cell number, 31 genes affected TZ cell number, and 32 genes affected cell number in both zones (Fig. 2a, Supplementary Figs. 4, 5 and Supplementary Data 5). While the remaining genes did not show statistically significant differences, large variations in the number of TZ and PZ nuclei were observed in many genes after RNAi (Supplementary Figs. 4, 5 and Supplementary Data 5). Knockdown of all major matrisome gene families, including metalloendopeptidases, cathepsins, serine

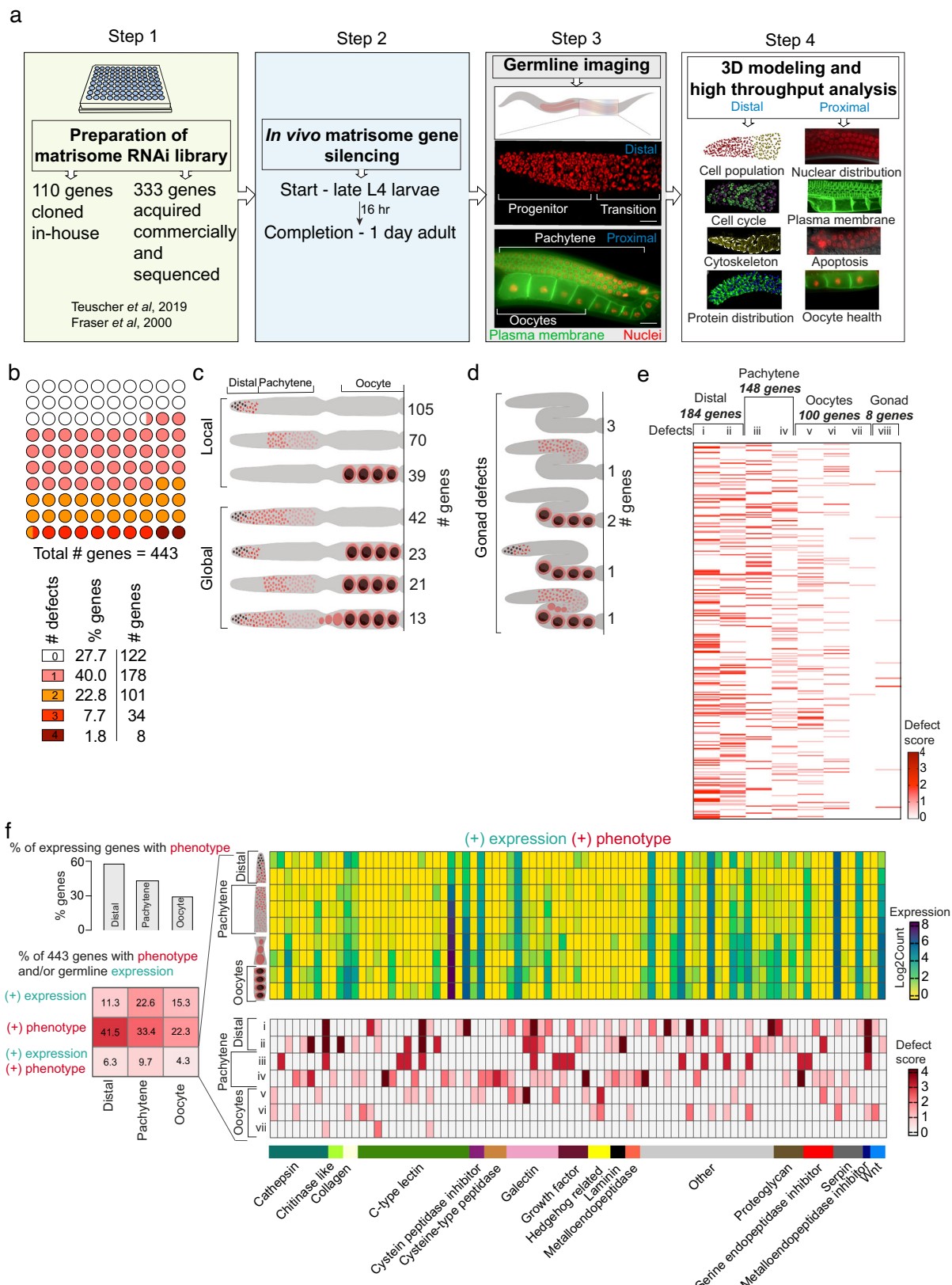

proteases, growth factors, laminins, and proteoglycans, reduced distal germ cell number (Supplementary Fig. 6 and Supplementary Data 5). Maintenance of distal germ cell number is controlled by proliferation of germ cells in the PZ. A cohort of germ cells within the PZ act as stem cells that undergo mitotic division to self-renew[32]. We hypothesized that a possible reason for changes in PZ cell number is altered mitotic cell cycle or premature departure from the mitotic cycle to enter meiosis. To classify known matrisome genes with roles in mitotic cell cycle regulation, we performed GO analysis in *C. elegans, Drosophila*, zebrafish, mice and humans (Supplementary Data 5). Using the mitotic cell cycle GO term (GO:0000278), we found that orthologs of *apl-1* (amyloid beta precursor-like protein), *let-756* (fibroblast growth

**Fig. 1 | Profiling matrisome functions in the *C. elegans* germ line. a** Experimental protocol for short-term RNAi-mediated silencing of matrisome genes and analysis of germline phenotypes. **b** Number of defects observed for each matrisome knockdown (0–4 phenotypes/per gene). Location of germline defects following RNAi-mediated gene silencing (distal, pachytene, oocyte regions (**c**) and/or gonad (**d**). Local defects identified at one location, and global defects at multiple germline regions or caused defects in gonad morphology. Number of genes generating the indicated defect after RNAi is shown. **e** Heatmap showing the number (bands/lane) and severity (defect score) of phenotypes after matrisome gene silencing. Each band in lanes represents a single gene. Defects are labeled as, i–PZ cell number, ii–TZ cell number, iii–multinucleated cell at pachytene, iv–apoptosis at late pachytene, v–defective oocyte morphology, vi–oocyte blebbing, vii–multinucleated

oocytes, viii–gonad defects. Defect score shows the severity of the defect. **f** Correlation analysis of germline expression pattern and phenotypes of matrisome genes. Bar graph shows percentage of expressed genes at a specific germline region (distal, pachytene or oocyte) that generate RNAi knockdown phenotypes at the same region. Table showing the percentage of total genes with phenotypes and/or expression in a specific region. Heatmaps show the germline expressed matrisome genes that generate at least one phenotype after knockdown in the region of expression. Top heat map–expression level. Bottom heat map–specific defects. Germline schematic shows the location of expression. Defects are numbered as in (**e**). Supplementary data associated to Fig. 1–Supplementary Figs. 1 and 2, and Supplementary Data 1, 3 and 4.

factor), and *ketn-1* (myotilin) are associated with mitotic cell cycle (GO:0000278) (Supplementary Data 5). Next, we investigated potential indirect links between our experimentally validated matrisome genes controlling PZ cell number and known genes involved in the mitotic cell cycle. We performed STRING (Search Tool for the Retrieval of Interacting Genes/Proteins) analysis of the combined list of our experimentally validated genes to assess the likelihood that they are indirectly associated with the mitotic cell cycle (Fig. 2b and Supplementary Data 5). This analysis indeed revealed a possibility that matrisome genes indirectly control the cell cycle. To explore this further, we selected 23 genes that significantly reduced PZ cell number ($p < 0.0001$) to study the germ cell cycle (Fig. 2a and Supplementary Data 5). Cell proliferation is determined by the time each nucleus spends in each cell cycle stage (Supplementary Fig. 6)[35–37]. Therefore, to study the effect of matrisome gene silencing on the cell cycle, we quantified cell proliferation (S phase stages of DNA synthesis) using 5-ethynyl-2′-deoxyuridine (EdU) staining (Fig. 2c)[38]. First, we established the number of nuclei at each stage of S phase in control germ lines (Supplementary Fig. 6). Using these control values, we calculated the normalized number of nuclei in S phase after silencing each matrisome gene (Fig. 2c, d and Supplementary Fig. 6). Silencing *zmp-5* (metalloendopeptidase), *let-756* (fibroblast growth factor), *lec-3* (galectin) and *apl-1* (amyloid β-precursor like protein) resulted in the reduction of total number of S phase cells (Fig. 2d and Supplementary Fig. 6). Analysis of the early, mid and late S phase revealed additional matrisome genes affecting S phase transitions (Fig. 2d and Supplementary Fig. 6). The number of nuclei in all stages of S phase was altered by silencing *apl-1* and *let-756* (Fig. 2d and Supplementary Fig. 6), whereas silencing *zmp-3* (metalloendopeptidase) affected only the early and mid-S phase stages (Fig. 2d and Supplementary Fig. 6). Finally, silencing *cri-2* (metalloendopeptidase inhibitor), *W07B8.4* (cathepsin), and *wrt-1* (hedgehog-related) altered the early S phase, and silencing *F32H5.1* (cathepsin), *timp-1* (metalloendopeptidase inhibitor), *skpo-2* (peroxidase), and *endu-1* (extracellular nuclease) affected late S phase (Fig. 2d and Supplementary Fig. 6). Together, these data confirm previously identified genes (*apl-1* and *let-756*) and reveal 10 matrisome genes that control the mitotic cell cycle (Supplementary Fig. 6)[39]. Further, we reveal specific matrisome factors that control S phase transitions that likely impacts PZ cell number (Fig. 2c, d and Supplementary Fig. 6)[40].

Access of germ cells to proteins required for cell cycle maintenance is important for cell cycle regulation[41]. The germline cytoskeleton maintains a "zig-zag" structure at the distal end that determines nuclei placement, protein distribution and somatic interactions in the PZ, and thus potentially affects proliferation (Supplementary Fig. 6)[34,41,42]. Further, actin generates mechanical forces that impact the mitotic cell cycle[43]. As several matrisome molecules are well known regulators of cytoskeletal organization[44], we hypothesized that changes in cytoskeletal structure at the distal germ line upon matrisome gene silencing may influence the germ cell cycle. Therefore, we used phalloidin staining to analyze actin cytoskeletal architecture of the distal germ line after silencing the 23 matrisome genes that

significantly change PZ cell number (Fig. 2e, f and Supplementary Fig. 6). We detected multiple distinct cytoskeletal changes following matrisome gene RNAi (Fig. 2e and Supplementary Fig. 6). Germline defects varied from simple loss of "zig-zag" configuration to total cytoskeletal disorganization (Supplementary Fig. 6). We found that silencing of 11 matrisome genes (*zmp-3, timp-1, cri-2, ctsa-4.2, W07B8.4, tag-290, W05B2.2, cof-2, apl-1, wrt-1* and *endu-1*) caused defective cytoskeletal structures in at least 50% of germ lines analyzed (Fig. 2f and Supplementary Fig. 6). Except for *tag-290, W05B2.2*, and *cof-2*, these genes are also required for correct cell cycle progression (Fig. 2d, f). To underpin these results, we performed network analysis between the genes with experimentally validated distal germline phenotype and genes associated with the broad GO term "actin cytoskeleton" (GO:0015629) (Supplementary Data 6). This identified 31 matrisome genes with experimentally validated defects at the distal region that are associated with the actin cytoskeleton genes (GO:0015629) (Supplementary Fig. 6), including five genes (*apl-1, cof-2, cri-2, tag-290*, and *wrt-1*) with verified cytoskeletal defects (Fig. 2f and Supplementary Fig. 6). Together, we reveal crucial roles for specific matrisome factors in maintaining germline structure that impacts germ cell behavior.

We speculated that the effects of matrisome factors on PZ cell behavior may be caused by changes in the distribution of germ cells and regulatory proteins due to cytoskeletal defects. To examine this, we studied the distribution of GLP-1/Notch Receptor—a critical regulator of germ cell proliferation in the PZ that responds to Notch ligands produced by the distal tip cell[7]. GLP-1 protein exhibits a gradient of expression in the distal germ line with highest expression at the distal tip (Fig. 2g and Supplementary Fig. 7) and is cleaved upon activation by Notch ligands[7,45,46]. The cleaved intracellular domain of GLP-1 (Notch IntraCellular Domain) then translocates to the nucleus to control transcription of target genes[46]. To study whether changes in germline cytoskeletal structure alter GLP-1 distribution from the distal end, we reconstructed 3D models of distal germ lines following nuclear staining and GLP-1 immunofluorescence (Fig. 2g and Supplementary Fig. 7). We analyzed GLP-1 distance of distribution from the distal tip and visible phenotypic differences in expression (Fig. 2g, h and Supplementary Fig. 7). This revealed six genes (*timp-1, R09F10.1, cof-2, wrt-1, zmp-3* and *cri-2*) that when silenced cause a significant difference in GLP-1 distribution from distal end of the germline. In addition, silencing nine genes (*timp-1, R09F10.1, cof-2, wrt-1, tag-290, ctsa-4.2, endu-1, lec-3* and *W07B8.4*) resulted in altered GLP-1 phenotype in ≥50% of germ lines. (Fig. 2h, Supplementary Fig. 7). Except for *lec-3* and *R09F10.1*, all these genes also exhibited cytoskeletal defects in the germ lines. Finally, we examined the distribution of nuclei in the PZ (Fig. 2i, j). To study nuclear distribution, we quantified the average distance between each nucleus and its three closest neighbors (Fig. 2i, j). The most significant gene knockdowns ($p < 0.0001$) showed a substantial reduction in cell number in the PZ. The distance between nuclei would be higher in those knockdowns compared to control if the PZ occupies the same volume. Alternatively, a closer distance suggests

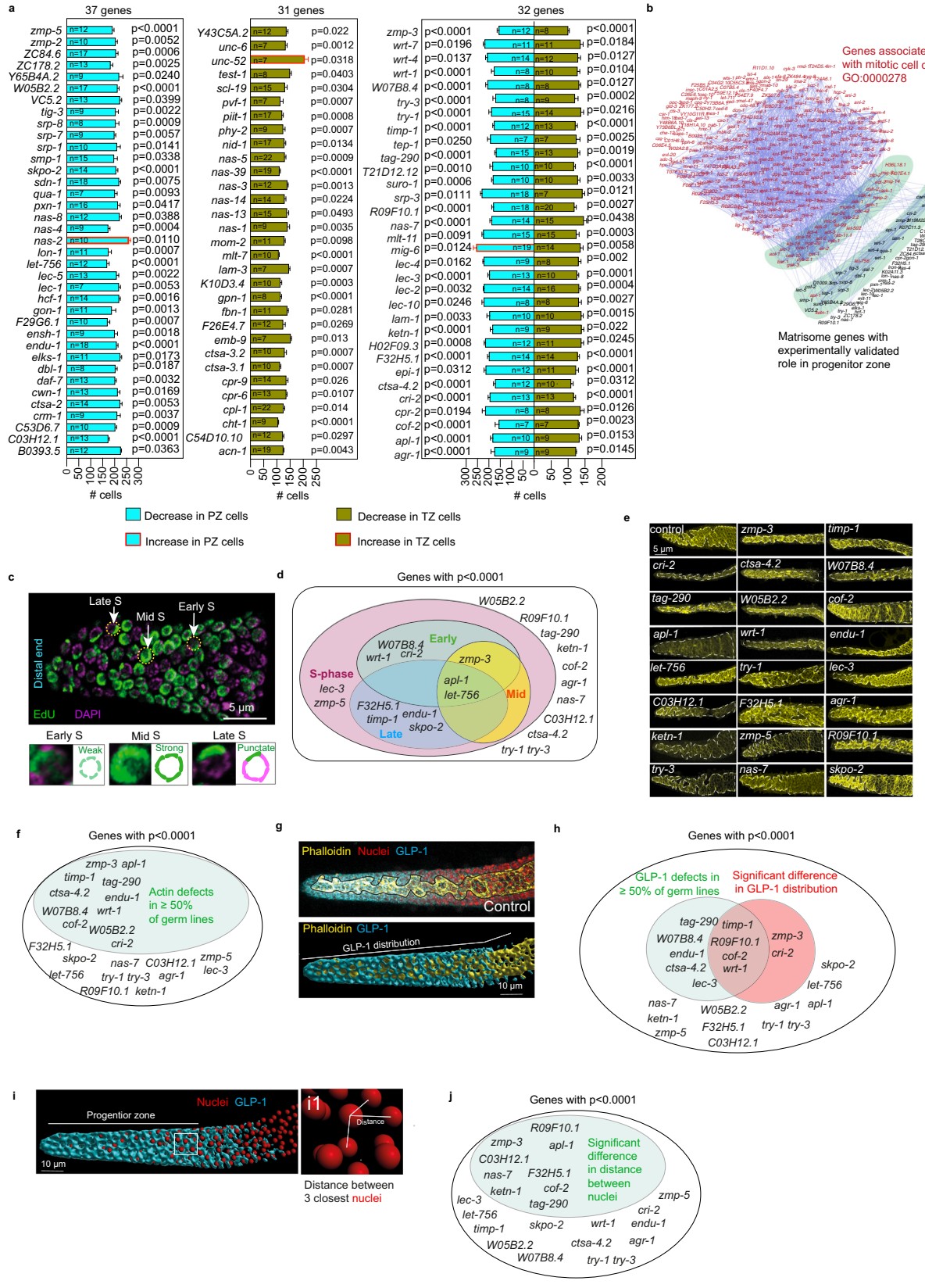

densely packed nuclei and a likely shorter PZ. The distribution of nuclei would also determine the proximity of each nucleus to regulatory proteins. The quantification of nuclear distribution after RNAi revealed that 9 genes (*zmp-3, apl-1, ketn-1, cof-2, nas-7, RO9F10.1, C03H5.1, tag-290*, and *F32H5.1*) had a significant difference in nuclear distribution (Fig. 2j). Together, our analysis of the cell cycle, cytoskeleton, and nuclear/protein distribution suggests that specific matrisome factors control progenitor germ cell behavior through multiple molecular and structural mechanisms (Fig. 2 and Supplementary Figs. 6, 7).

**Fig. 2 | The matrisome controls germ cell number and distal germline structure. a** Bar graphs showing the number of cells within the progenitor zone (PZ−blue) and transition zone (TZ−red) after RNAi. 37 genes affect PZ, 31 affect TZ, and 32 affect both regions. Data are presented as mean values ± SEM. Statistical significance and *p* values were calculated with respect to control for each RNAi (see Supplementary Figs. 4, 5 and Supplementary Data 5 for details, control values and data distribution for each experiment). **b** Network showing STRING analysis of association between matrisome genes with validated PZ function and known genes controlling the mitotic cell cycle. Red letter = genes associated with mitotic cell cycle (GO:0000278). Green shade = Genes have direct association in the STRING network. **c** Representative images of EdU and DAPI (pink) staining showing S phase stages. n per gene knockdown ≥ 6. Scale bar = 5 μm. **d** Venn diagram showing the most significant matrisome genes for PZ cell number difference (*p* < 0.0001 based on unpaired *t* test (for groups of 2) or ordinary one-way ANOVA (for groups of >2)) that control the cell cycle. Pink−total S phase, green−early S phase, yellow−mid S phase and blue−late S phase. **e** Micrographs showing cytoskeletal structure (phalloidin in yellow) at the distal end. Scale bar = 5 μm. n per gene knockdown ≥5.

**f** Diagram showing the most significant matrisome genes for PZ cell number difference (*p* < 0.0001 based on unpaired *t* test (for groups of 2) or ordinary one-way ANOVA (for groups of >2)) that affect cytoskeletal structure. Green−Gene knockdowns with ≥50% germ lines with defective cytoskeleton. **g** 3D reconstruction of endogenous GLP-1 (blue) and phalloidin (yellow) in the distal germ line. GLP-1 protein distribution and staining phenotype were quantified. Scale bar = 10 μm. n per gene knockdown ≥5. **h** Diagram showing the most significant matrisome genes for PZ cell number difference (*p* < 0.0001 based on unpaired *t*-test (for groups of 2) or ordinary one-way ANOVA (for groups of >2)) and their effect on GLP-1. Green circle- Genes showed an impact on GLP-1 phenotype in ≥50% germ lines. Red circle-Genes showed significant difference in GLP-1 distribution length. n per gene knockdown ≥5. **i** 3D reconstruction nuclei distal germline (red). Scale bar = 10 μm. i1−The distances between three neighboring nuclei to each nucleus in PZ. **j** Gene knockdowns with a statistically significant difference in nuclear distribution. *n* ≥ 5. Supplementary data associated to Fig. 2−Supplementary Figs. 2, 4, 5 and Supplementary Data 5.

## Specific matrisome components control meiotic cell behavior and oocyte development

As germ cells move distally from the PZ, they enter meiotic prophase I and later diakinesis prior to becoming gametes. *C. elegans* germline nuclei are partially enclosed with membrane, while sharing cytoplasm. Previous reports have shown that failure in cleavage furrow formation and cytokinesis of dividing stem cells or instability of germline structure cause the formation of multinucleated cells (MNCs)[20,41,47,48]. Since matrisome remodeling is thought to play a role in these processes, we investigated whether matrisome factors impact the occurrence of MNCs[48,49]. GO annotation using orthologs of *C. elegans* matrisome genes revealed that only annexin and hemicentin are known to be associated with cytokinesis (GO:0000910), cleavage furrow (GO:0032154), cleavage furrow formation (GO:0036089) or cellularization of cleavage furrow (GO:0110070) (Supplementary Data 7). Using a transgenic strain co-expressing fluorescent markers targeting nuclei (mCherry-histone H2B) and plasma membrane (GFP fusion that binds PI4, 5P$_2$), we quantified the number of MNCs per germ line (Fig. 3a and Supplementary Fig. 2)[50]. We found that 11% of control germ lines contain MNCs (Supplementary Fig. 8). We considered matrisome gene knockdown significant if we observed 55% (five times that of control) or more germ lines with MNCs. Using this criterion, we identified 21 matrisome gene knockdowns that increased the penetrance of MNCs (Fig. 3b and Supplementary Data 8). This includes proteoglycans (*C48E7.6*) metalloendopeptidases and their inhibitors (*acn-1, adt-3, nas-13* and *cri-2*), cathepsins (*asp-4, cpr-2* and *F32H5.1*), growth factors (*tig-2* and *tig-3*), hedgehog ligands (*wrt-9*), galectins (*lec-3, −9* and *−10*), semaphorin (*smp-1*), serine endopeptidase inhibitors (*ced-1* and *ZC84.1*), cysteine rich secretory protein (*scl-19*) and others (*lon-1, mua-3* and *hse-5*) (Fig. 3b). However, short-term RNAi of annexins (*nex-1, −2, −3 and −4*) and hemicentin (*him-4*) was insufficient to induce MNC, which contrasts previous reports in null mutant animals or long-term RNAi[51,52]. While we identified 21 matrisome genes that prevent MNC, the expressivity of these defects varied (Fig. 3c and Supplementary Data 8). For example, silencing *lec-10*, encoding a galectin, resulted in one MNC per germ line. In contrast, silencing *asp-4*, encoding a cathepsin, caused up to 12 MNCs in half of the germ lines analyzed (Fig. 3c and Supplementary Data 8). Taken together, we identified multiple matrisome factors that potentially control cytokinesis, cleavage furrow formation or germline structure stability to ensure that each germ cell contains one nucleus.

The *C. elegans* germ line is a self-contained tissue that can signal between distal, proximal and oocyte regions[7,10,53,54]. We hypothesized that defects observed in distal and pachytene regions would result in oocyte defects[9,10,48]. To study oocyte health, we analyzed nuclear morphology, membrane blebbing, and multinucleation in the three oocytes adjacent to the spermatheca (Fig. 3d and Supplementary

Fig. 2). First, we determined that defective oocyte nuclei, multinucleated oocytes, and oocyte blebbing were rare defects (~5% or less) in control animals (Supplementary Fig. 8). We classified a gene RNAi significant if it caused >25% penetrance (five times that of control) in oocyte defects (Fig. 3e–g). Using this benchmark, we identified 52 matrisome gene knockdowns that cause oocyte defects (Supplementary Data 8), with four genes (*asp-3, wrt-2, srp-1*, and *H02F09.3*) exhibiting multiple oocyte defects (nuclear morphology and blebbing) (Fig. 3e, f). The most common defect was defective nuclear morphology, followed by oocyte blebbing (Fig. 3e, f). Only knockdown of a cathepsin (*F32H5.1*) resulted in multinucleated oocytes (Fig. 3g). Analysis of oocyte defect severity following matrisome gene knockdown showed little difference between each knockdown (Fig. 3h–j). Except for *zmp-3* (metalloendopeptidase) in which all three oocytes were defective, all genes showed one (23 genes) or two (9 genes) oocytes with irregular nuclear morphology (Fig. 3h). Additionally, oocyte blebbing and multinucleation were only observed in one oocyte per germ line after RNAi (Fig. 3i, j). Taken together, our data suggest that specific matrisome factors control meiosis and oocyte formation, thus extending the role of the matrisome to the proximal germ line.

We found that only six out of the 21 genes that showed MNCs resulted in oocyte defects (Fig. 3k). Previous reports suggest that the MNCs are removed by physiological apoptosis in the germ line before becoming oocytes[55]. Except for *F32H5.1*, none of the gene knockdowns causing MNCs produced multinucleated oocytes. In addition, none of the gene knockdowns except for *zmp-3, cri-2 and F32H5.1* that exhibited distal cell cycle and structural defects also showed oocyte defects (Figs. 2d, f and 3l). These data suggest apoptotic clearance of defective germ cells prior to gamete generation[56]. To examine this, we quantified the normalized number of apoptotic cells per germ line compared with the control. We identified 51 gene knockdowns (26 upregulated and 25 downregulated) that significantly altered apoptosis. However, 20 out of 51 genes showed impaired apoptosis independent of detectable germline defects prior to apoptosis (Fig. 3m and Supplementary Data 8). While the remaining 31 genes showed changes in apoptosis as well as other defects (Supplementary Data 1, 8), strong links between increased apoptosis and early germline defects were not identified.

## C-type lectins are essential for *C. elegans* germline function

CLECs are calcium-dependent carbohydrate binding proteins that are secreted to the extracellular matrix and can coordinate cell-extracellular matrix signaling[57,58]. CLEC domain containing proteins have diverse functions including immune response, apoptosis and cell adhesion[57]. Based on literature curation and orthology, the *C. elegans* matrisome is predicted to contain >250 CLECs[27]. This contrasts the human matrisome with 28 CLECs[59]. We systematically silenced the predicted *clec* genes and analyzed the germ line.

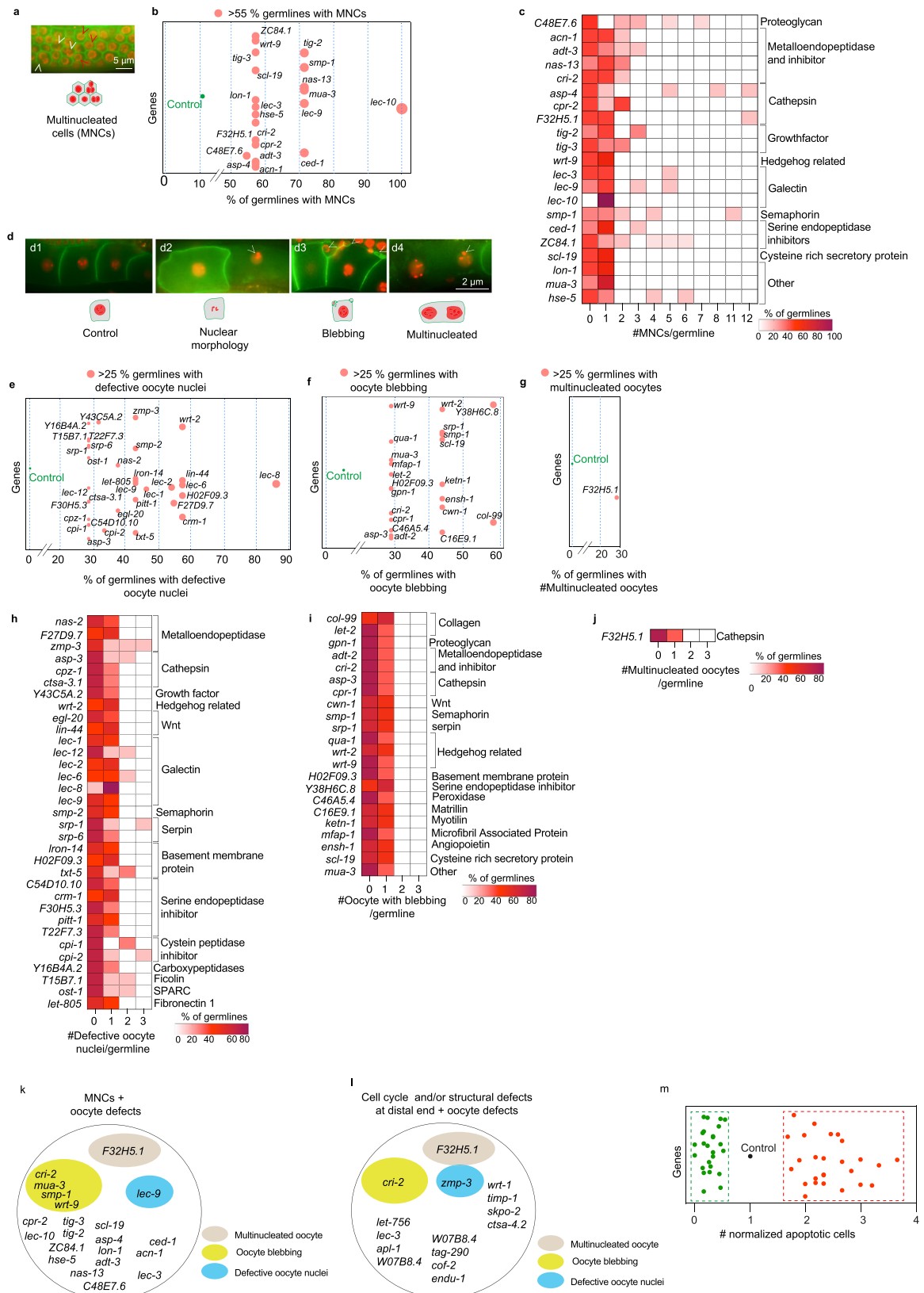

We identified 86 *clec* gene knockdowns that showed statistically significant differences in germ cell number at the distal end (38 in the PZ, 25 in the TZ, and 23 in both) (Fig. 4a, Supplementary Figs. 4, 5 and Supplementary Data 5, 9), and 34 gene knockdowns causing MNCs in the pachytene region (Fig. 4b and Supplementary Data 9). Silencing of many *clec*s resulted in germ lines with several MNCs (Fig. 4c)

suggesting a critical function for CLECs in germ cell behavior. However, GO coupled with STRING analysis did not reveal any association between CLECs (except *clec-59*, and −*64*) and genes involved in cytokinesis and cleavage furrows (Supplementary Data 9). Further, we identified 48 *clec* gene knockdowns leading to oocyte defects (Fig. 4d−f and Supplementary Data 9). While the severity of oocyte

**Fig. 3 | Matrisome control of meiotic germ cell behavior and oocyte health.**
**a** Micrograph showing multinucleated cells (MNCs) in the pachytene region of germ line. White arrowheads = MNCs. Red arrowheads = normal cells. Scale bar = 5 μm. n per gene knockdown ≥7. **b** Matrisome gene RNAi causing MNCs in >55% of germ lines (circle size proportional to the percentage). **c** Heatmap showing the severity of MNC formation. Each gene knockdown was plotted against the number of MNCs/germ line. Color (white to maroon) = % of germ cells with a particular number of MNCs/germ line. **d** Micrographs showing oocyte defects. **d1** = control, **d2** = oocytes with defective nuclei, **d3** = oocytes with blebbing, **d4** = multinucleated oocytes. Defects = white arrowheads. Scale bar = 2 μm. **e–g** Gene knockdowns resulted in oocyte defects in >25% of the germ lines (circle size proportional to the percentage). **h–j** Heatmaps showing severity of oocyte defects. Gene knockdowns

were plotted for defective oocyte nuclei, oocyte blebbing or multinucleated oocytes per germ line. Color (white to maroon) = % of germ lines with a particular number of oocytes with specific defects per germ line. **k** Correlation between MNCs and oocyte defects. 22 gene knockdowns resulted in MNCs, of which 6 had oocyte defects. **l** Correlation between cell cycle and structural defects at the distal end and oocyte defects. Of 15 genes required for cell cycle and distal end structure, only three showed oocyte defects. **m** Graph showing normalized germ line apoptosis. Each circle represents the normalized apoptosis after a gene silencing. Red/Green = Statistically significant increase/decrease in apoptosis, respectively. Black = Control. Supplementary data associated to Fig. 3–Supplementary Figs. 2, 8, and Supplementary Data 8.

defects depended on the CLEC type, only *clec-121, −150, −228,* and *-240* showed multiple oocyte defects (Fig. 4g–i and Supplementary Data 9). Finally, we found that silencing of 44 *clecs* decreased apoptosis in the germ line (Fig. 4j and Supplementary Data 9). Taken together, our data show the remarkable requirement of multiple CLECs for optimal germline functions. Some *clec* genes were previously shown to be important in *C. elegans* food intake. Since feeding and nutrition can directly impact the germ line, we tested selected *clecs* for their role in pharyngeal pumping[60]. Out of the eight *clec* genes analyzed, we only found one gene that regulates pharyngeal pumping, suggesting that the germline phenotypes observed are independent of feeding status (Supplementary Fig 9).

## Matrisome interactions and phenotypic mapping reveal a complex functional network

We identified 321 matrisome genes that play important roles in germ cells and/or oocytes (Supplementary Fig. 10). To identify the impact of each gene family on the germ line, we generated an integrated weighted average plot for each gene family. This shows how each family impacts the germ line considering the number of genes in the family causing phenotypes and phenotypic severity. Thus, a gene family will be considered as having high impact if it controls multiple germline regions with high severity, even if the family has a small number of genes. The majority of matrisome genes showed a comparable impact on the germ line. However, serine proteases, extracellular leucine rich repeats, and metalloendopeptidases inhibitors showed the highest impact (Fig. 5a). To summarize the functions of each gene family at specific germline locations, we determined the number and percentage of genes from each family that generated a phenotype (Fig. 5a). As expected, the CLEC family contributed the largest number of genes causing germline phenotypes, followed by metalloendopeptidases, serine endopeptidases inhibitors, cathepsins and galectins (Fig. 5a). The percentage revealed the proportion of genes (regardless of the total number in the family) from each family that resulted in a phenotype at a particular region of the germ line (Fig. 5a).

Matrix digestive enzymes, such as metalloendopeptidases, cathepsins, and serine proteases, also had profound effects on the germ line, suggesting a critical role of matrix remodeling in germ cell maintenance. The important core matrisome genes (collagen, laminin, nidogen, and proteoglycans), many of which are either substrates or regulators of matrix digestive enzymes, also showed germline defects[30,61]. Moreover, interactions between these genes may determine the matrisome functional output. We therefore examined how these putative interactions and phenotypes are linked. We performed STRING analysis of matrisome genes we found are important for germline development with curated interactions and mapped them according to the experimentally validated phenotypes (Fig. 5b and Supplementary Data 10). Curated interactions were generated only from validated experiments and functional databases with associated references. This analysis revealed 76 matrisome genes (of 321) with validated germline phenotypes can form 305 unique interactions (Supplementary Fig. 11 and Supplementary Data 10). Further, we

discovered 21 (of 76) matrisome molecules that interact with >10 other matrisome molecules with experimentally validated phenotypes (Supplementary Fig. 11 and Supplementary Data 10). Two inhibitors of metalloproteinases, *timp-1* and *cri-2*, showed the highest number of interactions with other matrisome genes with validated phenotypes, followed by the basement membrane component *F14B4.1* (Supplementary Fig. 11 and Supplementary Data 10). These interactions were generated based on curated data and thus do not guarantee phenotypic links.

To study the association between the interactions and phenotypes, we compared the identified phenotypes of each gene with the phenotypes of their matrisome interacting partners. We identified 60 out of 76 interacting matrisome genes (79%) that had at least one shared germline phenotype with their interacting partners following RNAi (Supplementary Fig. 11 and Supplementary Data 10). These 60 genes represented 157 (of total 305 interactions − 51%) interactions between matrisome genes (Supplementary Fig. 11 and Supplementary Data 10). We discuss the major groups here, and the complete list of interactions and phenotypes is detailed in Supplementary Data 10. Silencing the metalloproteinase inhibitors *cri-2* and *timp-1* resulted in four and two germline defects, respectively (Supplementary Fig. 11). The majority of their interacting partners, including their substrates (metalloendopeptidases), showed same defects upon knockdown (Supplementary Fig. 11)[62]. The basement membrane component *F14B4.1*, showed the second highest number of interactions (Supplementary Data 10). However, the only phenotype resulting from silencing *F14B4.1* was defects in gonadal morphology (Supplementary Fig. 11), which was not shared by any of the interactors. Silencing the matrix structural protein laminin (*epi-1, lam-1* and *lam-3*) caused defects at the distal end and an increase in apoptosis (Supplementary Fig. 11). These defects are shared by many of its major interactors, including proteoglycans (Supplementary Fig. 11). Proteoglycans are a group of receptors present both on the cell surface and in the extracellular matrix, where they act as receptors for a plethora of other matrisome molecules[61]. Our analysis verified that phenotypes after RNAi were similar between the proteoglycans (*agr-1, gpn-1, sdn-1* and *unc-52*) and their interactors (Supplementary Fig. 11). Another group of proteoglycan ligands are growth factors, some of which can regulate germline signaling[63]. We found that the growth factors (*daf-7, dbl-1, let-756, tig-2* and *tig-3*) and their interactors, all control germ cell number, MNCs and apoptosis (Supplementary Fig. 11). We found that multiple serpins, a CD109 molecule (*tep-1*), and amyloid β-precursor like proteins (*apl-1*) also have large number of interaction partners among the matrisome genes (Supplementary Fig. 11). Interestingly, curated interactions between CLECs and other matrisome molecules are minimal, even though they are the largest predicted family of matrisome molecules in *C. elegans* (Supplementary Data 10). Finally, we analyzed the functional pathways controlled by the genes in the network using the Reactome database[64]. Overrepresentation analysis of comparable mammalian orthologs suggest that 44 unique matrisome genes contribute to 170 signaling pathways. Among these, 15 matrisome genes were found in four pathways with a high confidence level

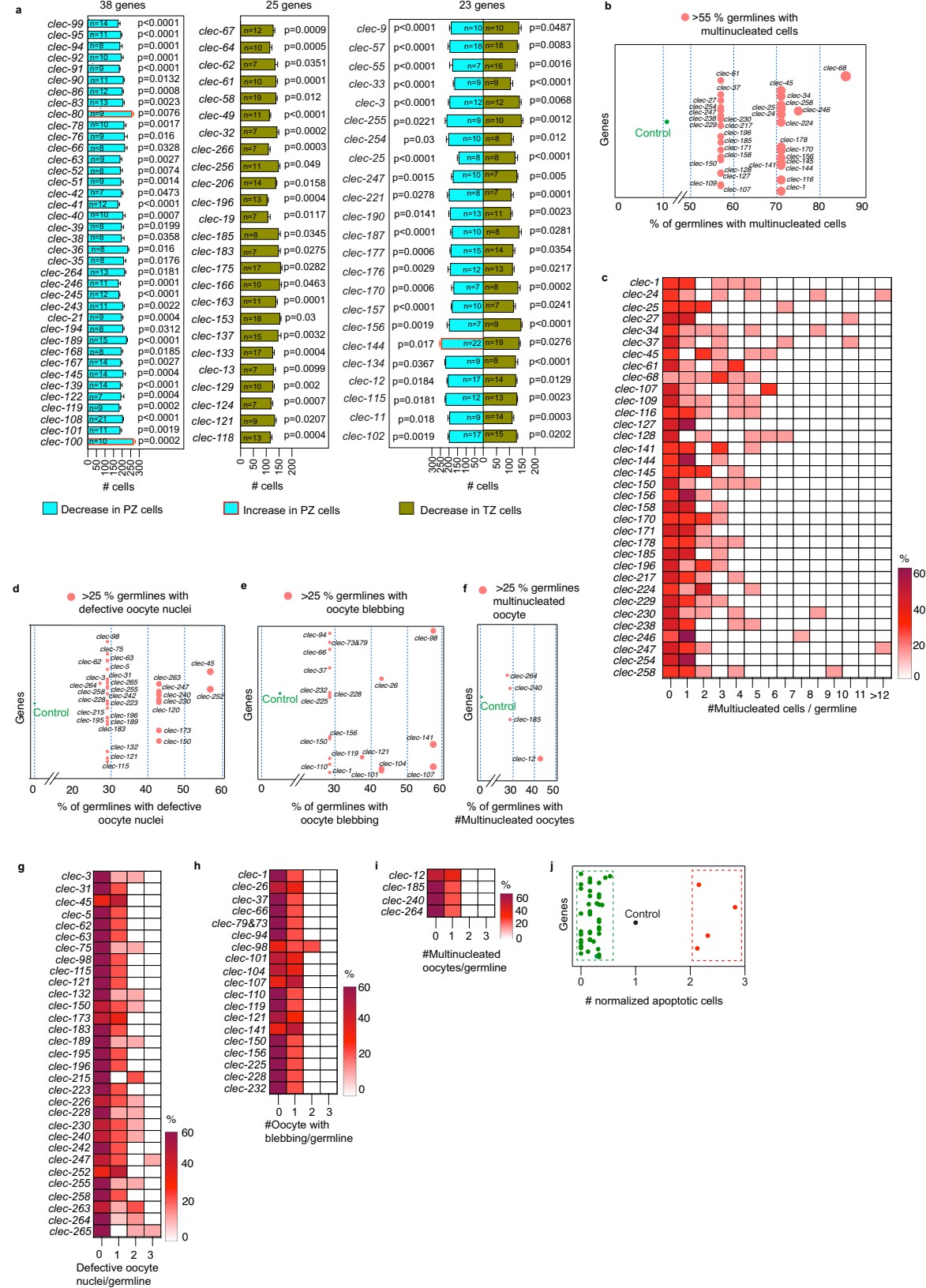

(Supplementary Fig. 12 and Supplementary Data 10). Overall, we identified a subset of genes that have physical and/or functional interactions in curated databases that produce specific local or global germline phenotypes after gene silencing. This posits an intricate network of molecules forms a matrisome landscape to control germ cell fate.

## Discussion

Our systematic silencing of matrisome genes in *C. elegans* provides insight into how the matrisome regulates germ cell fates in vivo. Our approach of short-term gene silencing allowed the acute requirement of matrisome factors to be assessed, without long-term secondary impacts on development. However, stable matrisome proteins that are

**Fig. 4 | C-type lectin silencing causes diverse germline defects. a** Bar graphs showing the number of cells within the PZ (blue) and TZ (red) after silencing *clec* genes. 38 *clecs* affect PZ, 25 affect TZ, and 23 affect both regions. Data are presented as mean values ± SEM. n per gene knockdown ≥7. Statistical significance and *p* values were calculated with respect to control for each RNAi (see Supplementary Fig 4, 5 and Supplementary Data 5 for details, control values, and data distribution for each experiment). **b** RNAi causing MNCs in >55% of germ lines (circle size proportional to the percentage). **c** Heatmap showing the severity of MNC formation. Each gene knockdown was plotted against the number of MNCs/germ line. Color (white to maroon) = % of germ cells with a particular number of MNCs/germ line cells. **d**–**f** *clec* knockdowns causing oocyte defects in >25% of germ lines. (circle size proportional to the percentage). **g**–**i** Heatmaps showing the severity of oocyte defects. Knockdown of each *clec* was plotted for defective oocyte nuclei, oocyte blebbing, or multinucleated oocytes per germ line. Color (white to maroon) = % of germ line cells with a particular number of oocytes with each defect per germ line. **j** Graph showing normalized germ line apoptosis. Each circle represents the normalized apoptosis after gene silencing compared with control germ lines. Red/Green = Statistically significant increase/decrease in apoptosis, respectively. Black = Control. Supplementary data associated to Fig. 4—Supplementary Figs. 2, 4, 5, 7, and Supplementary Data 10.

deposited prior to RNAi are unlikely to produce a phenotype after short-term gene silencing. The timing may also limit the transition of early defects to late defects (e. g. defects in the early meiosis leading to late defects such as apoptosis). In addition, several genes showed large variations after RNAi in phenotypes and failed to produce significant data. This could be due to variable efficiency or short exposure time of RNAi. Nevertheless, the analysis of phenotypes resulting from silencing matrisome genes reveals a complex functional network of molecules that control germ cell proliferation, differentiation, and apoptosis, to produce healthy oocytes[20]. The specific cellular/tissue origin of matrisome signaling requires further study, though several matrisome molecules are produced by the gonad itself. Some germline-associated basement membrane constituents that can signal to the distal niche are known[16,26]. However, germline associated basement membrane is not the only extracellular matrix present in worms. As the matrisome is produced by several tissues, it is likely that many of the phenotypes observed were a result of long-range signaling from other tissues. We found that changes in matrisome homeostasis can affect one or more germline processes. Cells in the PZ undergo mitotic cell division to maintain the germ cell population[65]. However, understanding how the matrisome controls the mitotic cell cycle is limited. We identified a large group of matrisome genes that control cell number at the distal end. These results explain the reason behind previously observed reduction in brood size and sterility in several matrisome gene mutant animals (Supplementary Data 2). We revealed an association between matrisome gene function and the mitotic germ cell cycle, which likely depends on the distal germline structure. Previous studies showed that germ cell overcrowding can alter germline actin structure[41]. However, our data show that a reduction in cell number is not always associated with a change in the actin structure. The role of matrix proteins in actin dynamics is well documented[66,67]. Further, actin plays critical roles maintaining mechanical forces in tissues, thus controlling cell behavior[67–69]. Therefore, it is likely that actin cytoskeletal structure, which is controlled by the matrisome, is a key factor in controlling cell behavior in the PZ. This highlights an additional layer of biochemical and biomechanical signaling in the *C. elegans* PZ that has not been explored previously. Similarly, we identified roles of matrisome genes in the pachytene region, where loss of matrisome gene function causes multinucleation of meiotic cells, indicating that some matrisome genes have roles in cytokinesis, cleavage furrow formation and/or stability of the germline structure. The most common phenotype we detected in oocytes was nuclear morphology changes, which resembles chromatin defects reported using the same strain[20]. Organisms possess mechanisms such as apoptosis that protect their germ line from inheritance of defective gametes. Our attempt to link early germ cell defects to apoptosis was not successful for two main reasons. First, a population of gene knockdowns showed oocyte defects, independent of early germline defects. Second, silencing of some genes caused changes in apoptosis independent of other defects. It is also likely that the acute gene silencing we employed did not allow compensatory mechanisms to take effect within the short timeframe.

Previous studies showed that loss of certain matrisome genes cause reproductive disorders in *C. elegans* and other organisms[7,30,70]. Our study provides the likely cellular causes underpinning these observations. Our collective data reveal the complex functions of matrisome factors. With a limited number of interactions in curated databases, we attempted to map these functions into an interaction network. Admittedly, several matrisome factors with experimentally validated functions require further interaction studies, which are beyond the scope of this study. Our data show that short-term RNAi-induced silencing of genes that encode extracellular matrix remodeling proteins had the greatest impact on germ cell development. This includes proteolytic enzymes (metalloendopeptidases and cathepsins) and their regulators that can remodel structural proteins such as collagens and laminins, after their deposition into the extracellular space[71]. Additionally, extracellular matrix modifications can alter ligand presentation to cells[72]. This suggests that matrisome modifications are continually required for controlling cellular fates in tissues. However, we found that silencing of substrates of the remodeling proteins resulted in weak germline defects. As mentioned previously, short-term RNAi may have limited impact on the substrates of remodeling proteins due to the abundance of the pre-deposited, stable proteins with slow turnover[73]. We found that growth factors and their likely receptors (proteoglycans) impact the distal to pachytene region of the germ line, aligning with their previously ascribed roles[6,7]. The current study revealed other functions for these molecules in controlling the proximal germ line. However, these molecules could function from the gonadal sheath, germline-associated basement membrane or elsewhere. We also identified widespread roles for galectins and CLECs in the germ line. While the number of predicted CLECs differs between human and *C. elegans* matrisomes, galectin composition is comparable[27,59]. Nevertheless, silencing of these genes produces a wide variety of germline defects. Both galectins and CLECs can bind to carbohydrates, thereby regulating glycosylated proteins[57,74]. Many animal proteins are glycosylated, and thus galectins and lectins may broadly regulate their function. This could be why loss of these genes affected germ cell behavior. However, we were unable to show this on the interaction-phenotype map because of the lack of verified studies. In addition, CLEC genes are important for broad immune responses and in selected metabolic signaling pathways, both of which can affect the germ line[60,74]. Taken together, our data reveal an exquisite requirement for a balance in the matrisome landscape to control germ cell fate and maintenance in the *C. elegans* germ line. Importantly, the lack of germline expression for many matrisome genes suggests that they regulate germ cell fate non-cell-autonomously. Finally, given that our analysis of matrisome gene function was after the early developmental stages were complete, our data reveal a continual requirement for remodeling of the matrisome landscape for faithful germ cell behavior and gamete generation.

## Methods
### Experimental model
*C. elegans* strains were maintained on Nematode Growth Medium plates and fed with OP50 *Escherichia coli* bacteria at 20 °C, unless

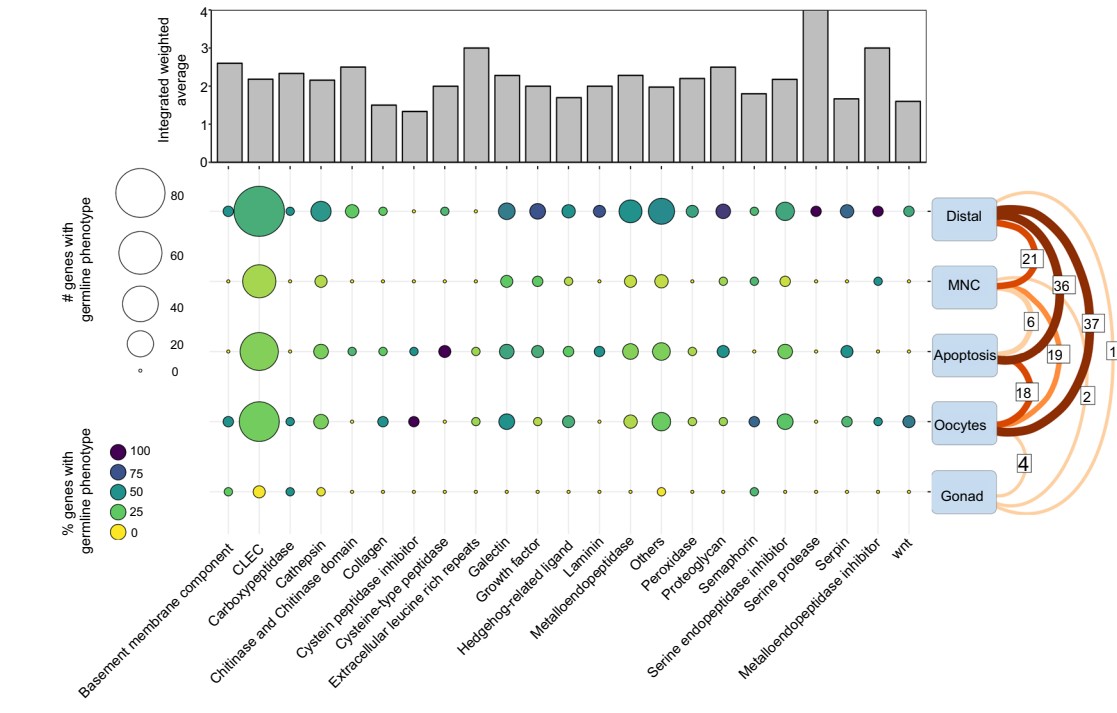

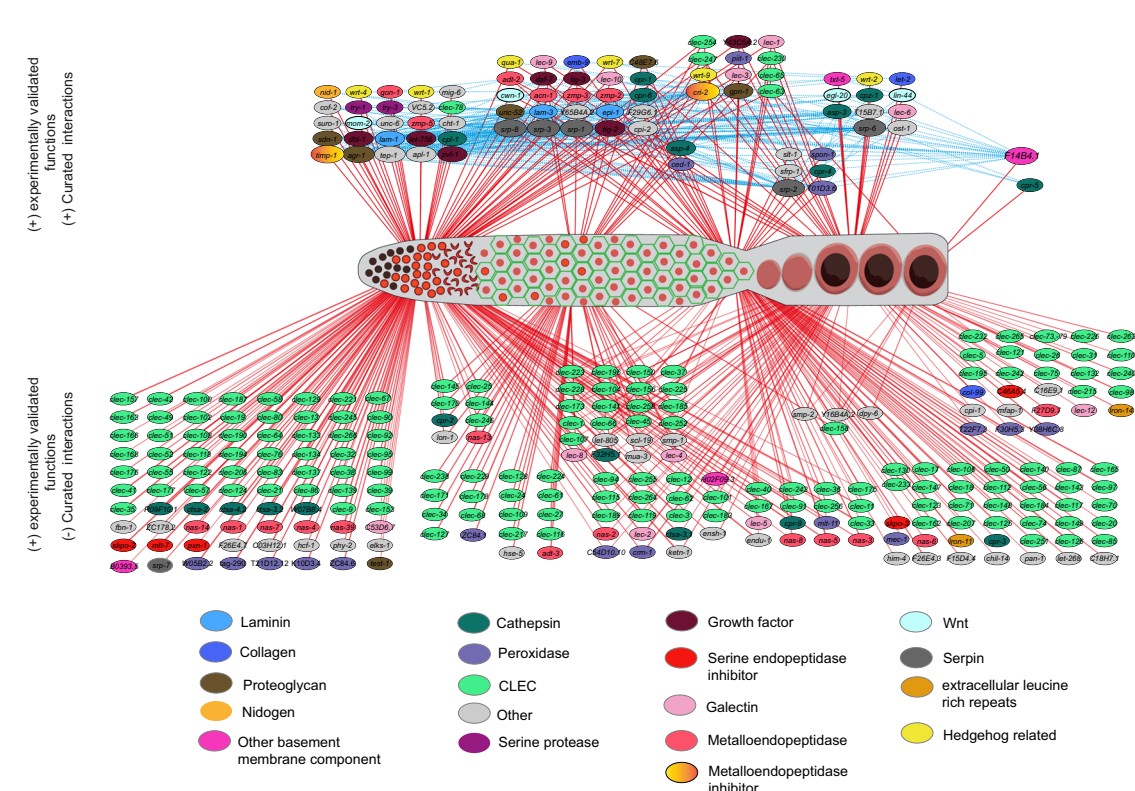

otherwise stated. All strains used in this study are listed in Supplementary Data 11.

### Reagents

Reagents used in this study are listed in Supplementary Data 11.

### Selection of matrisome genes

The matrisome genes were selected based on the previously published in silico characterization of the *C. elegans* matrisome[27]. We identified 443 conserved matrisome genes to study their role in germline development. We eliminated known pseudogenes from this study.

**Fig. 5 | Mapping matrisome interactions and associated germline functions.**
**a** Top panel−Bar graph showing the impact of each gene family in the germ line, calculated by weighting the phenotype average based on the number of genes causing the phenotype. Left axis−Integrated weighted average. 0−4 = low to high impact. Bottom panel−balloon plot showing the distribution of genes causing phenotypes in each region of the germ line. Size of the circle = number of genes showing a phenotype. Color of the circle = percentage of genes in each family showing a phenotype. The numbers on the right of the plot shows the number of genes that cause phenotypes in multiple regions. **b** Map showing the curated interactions between matrisome genes with experimentally verified phenotype. Top schematic−network showing all the genes regulating the germ line that have previously curated STRING interactions. Blue lines show interactions between genes, and the red lines show the location of germ line functional impact. Each gene family represented by node color. Gene families with less than two members indicated as other. Bottom schematic−Map of matrisome RNAi phenotypes, mainly constituting CLECs, that did not reveal interactions with other matrisome genes in validated databases. Supplementary data associated to Fig. 5−Supplementary Data 5 and 10.

Additionally, we eliminated 18 genes which we were not able to clone into the RNAi vector (Supplementary Data 11). The gene families and known phenotypes were generated as curated on Wormbase (https://wormbase.org//#012-34-5). The naming was adapted as shown in Teuscher et al. and Wormbase[27].

## Plasmid production for RNAi
RNAi plasmids for 333 genes were available commercially from Ahringer RNAi library (Supplementary Data 11)[28]. The plasmids were verified by Sanger sequencing using primer (RNAi F) listed in Supplementary Data 11. RNAi plasmids for 110 genes were cloned in house. The sequence for each gene was amplified using forward and reverse primers listed in Supplementary Data 11. HindIII sites were introduced at both ends while amplifying the gene sequence. The empty vector (L4440) and amplified product were digested using HindIII enzyme. The amplified gene was ligated into the vector using DNA ligase enzyme. The plasmids with verified sequences were transformed into HT115(DE3) *E. coli* bacteria for RNAi experiments.

## RNA interference experiments
HT115(DE3) *E. coli* bacteria expressing RNAi plasmids for specific genes or an empty vector (L4440) were grown in Luria Broth (LB) + Ampicillin (100 μg/ml) at 37 °C for 16 h. Saturated cultures of RNAi bacteria were seeded on RNAi plates supplemented with 1 mM IPTG and 50 μg/ml carbenicillin and dried for at least 24 h. L4 hermaphrodites were placed on RNAi plates and incubated for 16 h at 20 °C before proceeding with germline analysis. To confirm the efficiency of the protocol, another plate with RNAi against *pos-1* gene was used as a control. *pos-1* silencing results in embryonic lethality. The data from each experiment was accepted only if complete embryonic lethality was achieved in this *pos-1* control. The RNAi experiments were not performed in any particular order or family of genes. Initially, 8 different phenotypes were analyzed after RNAi, which are shown in Supplementary Fig. 2.

## Databases used in this study
Matrisome genes were selected from the database generated by Teuscher et al., which was created by comparing the human matrisome gene list (http://matrisome.org/)[27]. We excluded the genes listed as nematode-specific from further studies. GO analysis of matrisome functions was performed using WEB-based GEne SeT AnaLysis Toolkit (WEBGESTALT)[75]. The gene families and known phenotypes of matrisome genes were generated as curated on Wormbase or PubMed. The spatial expression of the genes was extracted from ref. 31. Gene lists for GO terms were generated using QuickGo−A web-based tool for GO searching[76]. The gene networks were created using data from STRING analysis (string-db, Version 11)[77]. Orthologs of *C. elegans* matrisome genes were generated using DRSC integrative ortholog prediction tool (DIOPT Version 9.1−https://www.flyrnai.org/cgi-bin/DRSC_orthologs.pl). DIOPT uses ortholog predictions made by Ensembl Compara, HomoloGene, Inparanoid, Isobase, OMA, orthoMCL, Phylome, RoundUp, and TreeFam. We included all genes that were found by at least one prediction program for initial ortholog analysis. To improve the accuracy of specific functional analysis by GO, we only used orthologs that showed moderate to high rank with a weighted score >5 from DIOPT. The signaling pathways were generated using Reactome

Knowledgebase (https://reactome.org), which predicts based on human pathways[78].

## Distal cell number analysis
Cell number at the PZ and TZ were analyzed by DAPI staining and 3D modeling (Supplementary Fig. 2). The genes were individually silenced as explained above. Germ lines were extruded from sedated worms and placed on a poly-L-lysine coated slide and quickly dipped in ice cold methanol for 15 s, and then fixed in 4% paraformaldehyde (PFA) in PBS for 20 min. Fixed germ lines were permeabilized twice in phosphate buffered saline (PBS, pH 7.4) containing 0.1% Tween 20 (PBST) for 10 min and blocked using 30% normal goat serum. The germ lines were incubated with 4′,6-diamidino-2-phenylindole (DAPI) for 2 h at room temperature. After staining, germ lines were washed twice with 0.1% PBST. Slides were mounted by applying a drop of Fluoroshield mounting media (Sigma) on the germ lines, followed by a coverslip. At least 7 germ lines from seven different animals per RNAi were imaged using confocal microscopes (Leica SP8 microscope at 63x objective, Olympus A1RHD microscope at 40× objective or Leica Stellaris microscope at 40x objective). The images were converted into 3D models by using spot function of Imaris 10.0 software (https://imaris.oxinst.com/) as described previously[7,34]. Briefly, the diameter of DAPI stained nuclei in the mitotic region were defined with a XY diameter (Example: For a 40× objective Leica SP8 at 1.7× zoom, $X$ and $Y = 2$–$2.2$ μm), while TZ nuclei is defined with same $XY$ diameter and additional $Z$ diameter (Example: For a 40× objective Leica SP8 with 1.7× zoom, $=1.5$ μm) to accommodate the change in shape. The diameters were decided according to the objective and microscope used (Note: This requires user optimization). The accuracy of diameter and the detection of the nuclei was visually confirmed using control RNAi samples for each experiment and the previously observed values as in a wild-type germ line were obtained[7]. The results were plotted as bar graphs and the statistical significance and $p$ value were calculated by comparing with the respective control values.

## Cell cycle analysis
S-phase analysis was performed by 5-ethynyl-2′-deoxyuridine (EdU) labeling via soaking. After RNAi feeding, worms were washed in a solution of M9 containing 0.01% Tween-20. An equal volume of EdU and M9 buffer were added to the wash solution to create a final EdU concentration of 250 μM. Worms were left in solution and rotated for 15 min before being placed on an unseeded agar plate and left to recover for 1−2 min. Germ lines were then extracted and fixed as described above. EdU-labeled cells were stained with a Click-iT Alexa Flour (488) EdU labeling kit (Invitrogen) by undergoing 2 × 30 min Click-iT reactions followed by 2 h DAPI staining. After staining, germ lines were washed twice with PBST. Slides were mounted by applying a drop of Fluoroshield mounting media (Sigma) on the germ lines, followed by a coverslip. EdU labeled germ lines (≥6 germ line from 6 different animals per RNAi) were imaged using Leica SP8 at 63× or Olympus A1RHD microscopes at 40× objective and analyzed using Imaris 10.0 software. The stages of cell cycle were identified based on intensity and shape of EdU staining (Fig. 2)[38]. By using Imaris, all the cells stained with EdU were marked automatically. The cells were manually observed to ensure correct Imaris-based identification. To

avoid the error in manual quantification of EdU staining intensity and variations between runs, intensity was detected using Imaris Software. The cells with highest intensity (top 30% of maximum intensity threshold) were identified as mid and late S phase in each germ line (Fig. 2)[38]. Then the two stages were distinguished manually by counting the cells in the late S-phase. Late S phase shows a punctate staining in the nuclei, which were counted and subtracted to obtain mid S phase cell number. The cells with low intensity (bottom 70% or less) were identified as early S-phase. The values were normalized against the values from control RNAi values obtained from multiple runs.

## Cytoskeleton analysis

For cytoskeletal analysis, germ lines fixed in 4 % PFA in PBS were permeabilized twice in PBS containing 0.1% tween-20 for 10 min or 0.1% triton X-100 for 10 min and stained using phalloidin (cytoskeleton) for 2 h. The germ lines (≥5 germ lines from 5 different animals per RNAi) were then imaged using Leica SP8 at 63× or Olympus A1RHD microscopes as explained above. The phenotypes were analyzed by observing the cytoskeletal structure at the distal end using Imaris 10.0 software. Gene knockdown was considered significant if the percentage of germ lines showing cytoskeletal defects was ≥50% after RNAi.

## GLP-1 and nuclei distribution analysis

Germ lines (≥5 germ lines from 5 different animals per RNAi) were extruded from the *glp-1(q1000[glp-1::4xV5]) III* strain after RNAi. This strain expresses GLP-1 with a V5-tag on its intracellular doamin[46]. The isolated germ lines were fixed and blocked as explained above. The slides were washed twice in PBST and incubated with 30% normal goat serum containing diluted antibody against V5-tag overnight at 4 °C. After incubation, germ lines were washed twice with PBST and incubated with fluorophore-conjugated secondary antibody and DAPI for 2 h. The slides were then washed, mounted and imaged, and 3D models of DAPI staining were generated using Imaris 10.0, as described above. By using an in-built tool in Imaris (distance between three closest neighbors), we quantified the distance between one nucleus and its three closest nuclei. The average of three distances for each nucleus was given as output from the software. The mean of these values was used to plot the graph and calculate statistical significance. To render a 3D model of GLP-1, surface function of Imaris was used. For each knockdown, respective control RNAi germ lines defined surface development parameters. The background, minimum, and maximum intensity threshold and detailing of surface were defined for the germ line using control RNAi. The values of these parameters depend on the microscope, objective, magnification, and staining quality the user defined. To avoid issues caused by variations in staining quality and intensity between runs, and subjective bias, each parameter was defined for a particular run using control samples. The defined parameters were saved to Imaris and recalled to generate 3D models for each gene knockdown. Detailed images are added to Supplementary Fig. 7, showing examples GLP-1 staining and developed 3D model of GLP-1. The changes visualized on the generated surface after RNAi were considered as phenotypic differences (Refer to Supplementary Fig. 7 for examples). A gene knockdown was considered significant if the percentage of germ lines showing visual variations in GLP-1 phenotype was 50% or more after RNAi. To analyze the distance of GLP-1 gradient distance from the distal end of the germ line, we used an in-built distance measurement tool in the software. This tool allowed us to draw lines along the length of the GLP-1 surface generated, accommodating turns and bends in the germ line.

## Analysis of MNCs, apoptosis, oocyte defects, and gonad defects

For analyzing the proximal phenotypes, *unc-119(ed3) III; ltIs37 IV; ltIs38* strain was used (Refer Supplementary Fig. 2 to see defect identification). This strain expresses mCherry tagged Histone 2B (nucleus) and

GFP fusion binding PI4, 5P$_2$ (plasma membrane), which allows observation of nuclei and the plasma membrane in live animals[20]. Germ lines (at least 7 germ lines from 7 different animals per RNAi) of live animals sedated on agarose pad using 0.01% levamisole were manually analyzed using Zeiss Axiocam microscope at 63x. Each germ line was analyzed for phenotypes listed in (Supplementary Fig. 2)[20]. The percentage of germ lines with MNCs and oocyte defects was calculated. If a gene knockdown showed defects 5 times more than control, it was identified as significant. To investigate the severity of MNCs and oocyte defects, the number of defects per germ line was calculated. Apoptosis was quantified by counting the nuclei with morphology as explained previously[20]. Briefly, the apoptotic cells were recognized by a condensed nuclear morphology and strong mCherry (Histone 2B) intensity in *unc-119(ed3) III; ltIs37 IV; ltIs38* strains and confirmed by differential interference contrast (DIC) microscopy (Refer Supplementary Fig. 2). Normalized apoptosis after knockdown was calculated by using values obtained from control RNAi from 302 germ lines. The gonad defects were calculated by manually counting the germ lines that do not maintain the U shape of gonad in the worm.

## Pharyngeal pumping assay

RNAi were performed for selected *clec* genes (with number of defects ranging between 1 and 3) as explained above. The pharyngeal pumping of the resultant 1-day old adult animals in plates containing food was counted for 20 s. Five worms were analyzed for each experiment and experiments were repeated three times.

## Gene Ontology analysis

GO search was performed using QuickGo–A web-based tool for GO searching[76]. The following terms were used in this study–Germ cell development (GO:0007281), gamete generation (GO:0007276), mitotic cell cycle (GO:0000278), actin cytoskeleton (GO:0015629), cytokinesis (GO:0000910), cleavage furrow (GO:0032154), cleavage furrow formation (GO:0036089), cellularization of cleavage furrow (GO:0110070).

## Generation of diagrams, bar graphs, bubble plots, heatmaps, and balloon plots

All figures and drawings within are created using Adobe Illustrator 2022. The heatmaps for Fig. 1f were constructed using Complex Heatmap package of R language. Other heatmaps were generated using GraphPad Prism 7.0. The balloon plot was visualized using ggplot2 package of R language[79]. Bubble plots for MNCs and apoptosis were generated using GraphPad Prism 7.0 and figures created using Adobe Illustrator.

## Generation of STRING network

The STRING network was constructed based information fetched from STRING database[77]. The network only included the experimentally verified and curated interactions. For Fig. 2b, the network was constructed from a combined list of two gene sets. One set of genes were associated with mitotic cell cycle and the second set of genes with experimentally verified distal phenotypes. The STRING network shows the association between these two data sets. Similarly, Supplementary Fig. 6k was constructed from genes associated cytoskeleton and with experimentally validated PZ. Finally, for Fig. 5b was constructed with experimentally verified matrisome gene list and the network was visualized using Cytoscape software (version 3.10.0)[80].

## Quantification and statistical analysis

Statistical analyses were performed in GraphPad Prism 7 using one-way analysis of variance (ANOVA) for comparison followed by a Dunnett's Multiple Comparison Test where applicable. Unpaired *t*-tests were performed if the comparison was for two conditions. For GLP-1 and nuclei distances Welch's *t*-test (2 groups), ordinary one-way ANOVA

and Brown-Forsythe and Welch ANOVA tests (>2 groups). For apoptosis, Welch's $t$-tests were used. Values are expressed as mean ± S.E. Differences with a $P < 0.05$ were considered significant. The integrated weighted average (Fig. 5a) was calculated for each family of matrisome genes according to the following:

$$\text{Integrated weighted average} = \frac{W1.A1 + \ldots + Wn.An}{W1 + \ldots + Wn} \quad (1)$$

A = average phenotype W = weight of the genes of the family in each region calculated as follow:

$$\text{Weight} = \frac{\text{Number of genes causing phenotype in each region}}{\text{Sum of genes causing phenotypes across all regions}} \quad (2)$$

### Statistics and reproducibility
The data shows a screen of 433 genes to study their impact on germ cells. For the initial analysis of 8 phenotypes, at least 7 germlines from 7 different animals were analyzed. The sample sizes for confirmatory experiments, cell cycle, cytoskeleton, pharyngeal activity, and protein distribution are added to individual experiments. Statistical data is provided with individual experiments. The investigators were not blinded to the source of the experimental samples and outcome assessment. The analysis was automated with predefined parameters wherever possible to avoid subjective variations.

### Reporting summary
Further information on research design is available in the Nature Portfolio Reporting Summary linked to this article.

## Data availability
The raw data generated in this study are provided in the Supplementary Information/Source Data file. Reagents and further information will be available upon request from the Gopal laboratory. Source data are provided with this paper.

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

## Acknowledgements

We thank Prof. John Couchman, Prof. Anders Malmstrom, and Prof. Gunilla Westergren-Thorsson for advice and comments on the manuscript. We thank Lisa Karlsson for generating schematics. Imaging for this project was performed at Monash Microimaging, Lund Bioimaging Center, and Lund Stem Cell Centre Imaging Facility. Some strains were provided by the *Caenorhabditis* Genetics Center (University of Minnesota), which is funded by the NIH Office of Research Infrastructure Programs (P40 OD010440). The gene family classification and functions were performed as curated in the Wormbase. This work was supported by the following grants. Swedish Research Council 2019-02020 (S.G.). The Crafoord Foundation 20200545 (S.G.). Cancerfonden (SG) 22 2125 Pj (S.G.). Royal Physiographic Society of Lund (S.G.). Franke och Margareta Bergqvists Stiftelse (S.G.). Australian Research Council DE190100174 (S.G.). National Health and Medical Research Council GNT1161439 (S.G.). Australian Research Council DP200103293 (R.P.). National Health and Medical Research Council GNT1105374 (R.P.). National Health and Medical Research Council GNT1137645 (R.P.). National Health and Medical Research Council Ideas Grant 2018825 (R.P.).

## Author contributions

S.G. and R.P. developed the concept. A.A., L.P., J.F., R.G., and S.G. developed the methodology. A.A., L.P., J.F., R.G., and S.G. performed the investigation. S.G. and R.P. acquired funding. Project administration was carried out by S.G. S.G. and R.P. supervised the project. S.G. wrote the original manuscript. S.G., R.P., A.A., L.P., and J.F. performed review & editing.

## Funding

## Competing interests

The authors declare no competing interests.
