## [Peer Review File · Nature Communications]

The matrisome landscape controlling *in vivo* germ cell fatesREVIEWER COMMENTS

Reviewer #1 (Remarks to the Author):

Using RNAi in *C. elegans*, Amran et al. systematically knocked down 443 conserved genes predicted to encode components of the extracellular matrix. Most of these knockdowns (>70%) caused germline defects. This study considerably expands the list of extracellular matrix genes involved in germline development in *C. elegans* because most of the tested genes have not been functionally characterized previously. The authors evaluated multiple phenotypes allowing them to assign genes into those regulating germline proliferation, apoptosis, oocyte morphology, etc. I found the study technically sound. I was concerned about presentation, although it is not easy to present these complex data simply.

I will comment on Figure 2 because I found it most challenging and because it had issues that reoccurred in other figures. The single purple dot in Figure 2a (mis-labeled as 2b), implies that there was only one control for all experiments. I presume this was not the case. Showing variability, which is likely considerable, could be useful. “Not significant” genes could have been caused by consistent lack of effect or by variable effects. These two categories may be meaningfully different. I suggest this should be displayed, possibly in the supplement, and discussed. I did not understand the point of color-coding. If genes were categorized as “progenitor” or “transition” based on overall phenotypes, wouldn't they preferentially affect numbers of cells in the PZ and TZ, respectively? Dot size and particularly opacity were hard to see, except in the most obvious cases. The Venn diagram looks strange since 39 isn't that much greater than 32. Plotting 2a differently and maybe as separate panels may help.

I did not understand the message of Figure 2b.

Figures 2c-g showcase how complex the phenotypes are. Even among the select 23 genes with defects in the # of PZ cells, there is a complex and variable pattern of effects on the S phase. Extracting these patterns from 2d-g was not easy. Wouldn't it be clearer to list genes putatively affecting different portions of the S phase under relevant images in 2c? The data currently in 2d-g could go into supplement or be presented in a way that would make it easier to see effects of genes on different portions of the S phase.

In Figures 2a, 2k, and 2m it might make sense to show only those genes that caused significant effects. This would focus analysis and discussion on possible biological insights rather than the number of genes tested.

I found Figures 3-5 generally easier to follow, although some elements in them have the same problems as in Figure 2.

Minor:

“...cell behavior analysis of >3500 germlines and >7 million germ cells” seems needlessly granular in the Abstract

I did not follow the point of lines 1-10 at the top of p. 5. Do these percentages really matter? I can see utility of briefly stating autonomous vs. non-autonomous, but am concerned whether we really know expression patterns of the putatively “non-autonomous” genes well enough to make a strong claim. I recommend shortening this long paragraph to the most essential statements and being explicit regarding limitations of such analyses.

Reviewer #2 (Remarks to the Author):

The manuscript of Amran et al. provides the first comprehensive examination of germline phenotypes caused by perturbation of ECM components. Short-term RNAi exposure was followed by several different methods (DAPI staining, EdU labeling, analysis of GLP-1 localization, DIC imaging, ...), to look at different stages of germ cell development. The authors employ various approaches to perform pathway analysis to link phenotypes to interacting protein partners. Overall, the data is of high quality and provides some of the first insight into the complexity of ECM contribution to germ line functions.

There are two limitations of the work: The first is that the study only relies on RNAi and does not present validation of phenotypes either with mutations of conditional AID alleles. While there are a few examples that show consistent phenotypes, there were also several that showed different results, leaving one to wonder. While it would be onerous to ask the authors to make such alleles for all of the genes, they could do this for a few of the genes with novel phenotypes or at a minimum they should acknowledge this more robustly and perhaps add a note somewhere in their tables or figures about which genes have known phenotypes or have phenotypes (such as embryonic lethality) that would preclude analysis by other methods. The second caveat with the work, that the authors acknowledge, is the exposure of RNAi for only 16 hours, which may not have been sufficiently long enough to reduce or eliminate function of stable ECM components. Consistent with this, the RNAi with the strongest phenotypes encode signaling molecules and modifying enzymes not core ECM structural proteins. The short time frame may also have prevented a subset of phenotypes from being observed, e.g. a meiotic defect may interfere with chromatin structure, but this may not be visible until later in the germ line. That said, the authors address this concern in the discussion (although this aspect could be lengthened a bit) and it does not detract from an otherwise ambitious study that moves the field forward and will be of broad interest.

Several issues that can readily be addressed would improve the manuscript:

- It would be helpful to see a fluorescent or DIC image of an example of each type of defect. This could be shown together in one supplemental image, but would really help the reader understand the breadth of phenotypes examined from the beginning.

- One general comment about data presentation. While I understand that the imaging of the anterior gonad, puts the distal end to the right, in general since the germ cells move distal to proximal, most papers put the distal end to the left and the oocytes to the right. This would flip the images in Fig 1c and d, make the graph in Figure 1e, read i – viii instead of reverse. I think this would help readers compare the results to other published studies. It would also reverse the image in Figure 5 and associated genes to read left to right from distal to proximal (younger to more developed).

- The gonad tube defects are confusing since the RNAi exposure began in L4 and persisted for only 16 hours. At these times, only the most distal part of the gonad tube is not fully elongated. So what is the nature of the defect in these cases?

- It is unclear which of the subset of genes from Fig 1E are being shown in Figure 1F.

- Timmons and Fire should be referenced for RNAi feeding. Fraser made the first libraries, but Timmons developed the protocol. Please reference accordingly.

- Page 5 lines 1-2 suggests that multiple expression datasets were examined for germ line expression but only Tzur 2018 is referenced. Were there other datasets that were examined? If not, why was this the one chosen? These expression data presumably include germ line (i.e. germ cells and DTC and sheath cells). Please make this clear in the text.

- I am confused by the yellow bar along the left edge of figure 2H since the control is at the top, what is the bar marking? It would be nice to specifically label the images based on how you describe them in the text, e.g. “completely disorganized”. Are one of the images supposed to represent that class?

- It is curious in Figure 3f that mutant RNAis cluster with ~30% and ~43% blebbing. Is there something unique about these values?

- The nuclear morphology defect in Figure 3D.d2 looks like a defective chromatin, possibly due to DNA fusions (e.g. mre-11 mutants would look like this). The nucleus, however, is otherwise normal (round, etc). Are all of this class of defects associated with aberrant chromatin morphology?

- In Fig 3D, d3, is it possible that blebbing is not actually from the oocytes but rather defective engulfment of apoptotic nuclei in the proximal gonad just above these oocytes?

- Figure 5a should have distal at the top and oocytes at the bottom for consistency between figures (e.g. Figure 1f)

- One caveat to the statement that “Our data show that silencing genes that encode extracellular matrix remodelers had the greatest impact on germ cell development.” is that the RNAi might have been most effective in reducing the function of this class of proteins, which they address below. Perhaps instead state “Our data show that short-term RNAi-induced silencing of genes that encode extracellular matrix remodelers ...”

- When it is stated that 7 germ lines/RNAi were analyzed, can you confirm that these are from 7 different animals or could 2 germ lines have been from the same worm?

Textual and Figure corrections

In the text, the authors should note that they have no idea which cells are actually laying down the ECM components they are studying. In some cases, ECM components, e.g. hemicentrin, are made by cells at a distance and are transported to the gonad.

I cannot read the text in Figure S1g.

Germline should be two words when used as a noun

references 14-16 are an odd choice to support the statement "as the cells move proximally, they enter meiosis in the transition zone". I might instead reference the WormBook reviews on germ line development and possibly meiosis. Similarly ref 18 is not the earliest paper to describe differentiation from pachytene into sperm and oocytes.

p. 3 line 22 "The role of extracellular signaling remains unclear"¹⁷. I think you need a clarification here between cell-cell signaling vs ECM-cell signaling since that is, I think, what you are really getting at. There are excellent descriptions of cell-cell signaling between the DTC and the germ cells to control proliferation and communication between sheath cells and germ cells via junctional molecules.

p. 3 line 27 "RNA-mediated interference" should just be "RNA interference (RNAi)"

Figure S2C "Time spend in each S phase" should read "Time spent..."

p. 5 lines 24-25 should also include reference to Crittenden and Kimble for size of the distal mitotic region.

p. 8 line 16, technically cells enter leptotene/zygotene before pachytene. might even be better to say "they enter meiotic prophase I"

p. 8 line 17-18 not quite sure what is meant by: "During meiotic pachytene, germ cell nuclei organize within plasma membrane boundaries through cleavage furrow formation and cytokinesis^{26,35}". The nuclei in both the mitotic and meiotic region are ensheathed by membrane, and open only to the shared cytoplasmic in the rachis. Cellularization happens not in pachytene but begins at the onset of diplotene and completed as nuclei reach the diakinesis stage.

IN extended figure 3 how are you defining multinucleated cells vs multinucleated oocytes?

Figure legend extended data 3 "11% of germlines with showed MNCs after control RNA" should read "Eleven percent of control RNAi germ lines showed MNCs"

“However, short-term RNAi of 35 annexins (nex-1, -2, -3 and -4) and hemicentin (him-4) was insufficient to induce MNC, which contrasts a previous report in null mutant animals³⁹.” hemicentrin is referenced, but nex-1 – 4 are not and should be.

p. 9, lines 8-9 The *C. elegans* germline is self-contained tissue that is capable of cross-talk between distal, proximal and oocyte regions^{12,17}. Ref. 17 here is less appropriate but there should be references to the work from the Greenstein laboratory.

The statement: p. 9 lines 9-10 that “We hypothesized that defects observed in distal and pachytene 10 regions would result in unhealthy oocytes.” is based on many papers that show that upstream mitotic zone and meiotic defects lead to oocyte defects. One example is the him-4 paper previously cites, but there are many others. And technically, this is not true crosstalk (line 8) between the regions, it is simply a developmental progression.

What does GO:0015629 term refer to?

P. 9, Line 29, there is an extra space before Ref 40.

p. 10, lines 2-3. Please briefly mention how you examined apoptotic nuclei.

The abbreviation for CLECs is introduced on p. 11, but is used extended data 3 legend which is discussed prior. Please spell out C-type lectin in that figure legend as well.

p, 11 line 29 “To study...” should start a new paragraph for ease of reading.

p 12 line 32 should read “germ cell fates” not “fate”

p. 13, line 15-16 “It is likely short time of gene silencing did not allow other mechanisms take into effect.” is not a sentence.

p. 13, last line. Hyphenate “hedgehog-related”

p. 15 line 3 italicize “*E. coli*”

p. 15 line 5, please define “RNAi plates” DO these have IPTG to induce the dsRNA? What concentrations.

p. 15 line 33, Edu should be Ed and in multiple other places in that paragraph.

p, 16 line 23 “cytoskeleton” should be “cytoskeletal”

p. 6 line 29. Mistaken reference and wrong format, “Germlines (7 germlines per RNAi) were extruded from the *glp-1::v5(q1000)* strain after RNAi. This strain expresses GLP-1 with a V5-tag (29).”

p. 17 line 25 “condensed nuclei morphology” should be “nuclear”
Extended figure 4 legend, unclear first sentence.

Germ tube vs. germtube. Both are used.

Text in extended fig 5a is hard to read even blown up 250% on my screen.

Fig 5 legend “network showing all the genes regulating the germline and have curated interactors.” do you mean “..and that have curated..”

Reviewer #3 (Remarks to the Author):

The authors use late-larval feeding RNAi to 443 extracellular matrix (“matrisome”) genes in a strain with fluorescently labeled germline nuclei and membranes. They assess 8 different germline phenotypes relating to positional defects along the gonad axis and observe both apparently cell-autonomous and non-cell autonomous effects of knockdown. Site of action is inferred from overlap between published transcriptomics datasets reporting location of expression and new observed positional RNAi-induced defects in the germline (site of action not experimentally determined). Surprisingly (to this reviewer, at least), the bulk of the matrisome genes with RNAi that causes germline defects belong to the class of C-type lectins.

In addition to the germline morphology assessment, the authors performed an EdU experiment to assess germ cell cycle perturbation after RNAi to genes that cause apparent distal proliferative zone defects. They assess the cytoskeleton (as a likely target of matrisome factors) by phalloidin staining and examine an existing GLP-1::V5 tagged Notch receptor localization in these RNAi conditions, though I have concerns about their interpretations of these results (concerns 1 and 2, below). They examine more proximal phenotypes like multinucleate germ cells, changes in apoptosis, and oocyte defects as well. They do pathway analyses to identify functional classes of genes.

My assessment: Overall, this study has generated a list of genes of interest and preliminary investigations into their likely locations of action to affect germline biology. These candidates seem to be the major take-home of the study rather than any particular biological conclusion. The study is methodologically similar to Green et al., 2011, in *Cell* (see Literature, below), with a focus on a different group of genes. I have several concerns about the interpretations of results, along with a few additional notes including concerns about deficits in scholarship.

Concern 1:

The zig-zag morphology of the distal germline was discovered by Seidel et al. (2018) to be a consequence of germ cell crowding—germline growth through proliferation while packed in the gonad tube caused folds to form in the germline. Therefore, the direction of causality inferred in the Results section running through bottom of 6-top of 7 seems backward:

P6 35 ...As matrisome molecules

P7 1 play crucial roles in cytoskeletal organization³³, we hypothesized that changes in cytoskeletal
2 structure at the distal end of the germline upon matrisome gene knockdown could influence the
3 germ cell cycle.

Page 13 line 7 in the Discussion repeats this mistaken direction of causation:

6 the cell cycle of mitotic germ cells, which likely
7 depend on the structure of the distal germline.

The authors measure proliferation defects, which (according to the Seidel 2018 model) predict less
folding of the distal germline. Why conclude instead that the folding of the germline affects germ cell
proliferation? These results need better integration with existing knowledge of germline biology.

Concern 2:

This framework for assessing GLP-1 localization described on Page 7 is problematic:

25 ...GLP-1 is also localized around nuclei, forming a
26 boundary (Fig. 2j; Extended data fig. 2h)^{12,34}.

GLP-1 is a Notch receptor; it is a transmembrane protein that (when activated by extracellular ligand
binding) cleaves the Notch intracellular domain (NICD), which then translocates to the nucleus and acts
as a cofactor with transcriptional regulatory Notch signaling effectors. Depending on the method of
visualization, different domains of the GLP-1 protein can be detected.

According to Sorensen et al. (2020), the position of the V5 tag in the GLP-1::V5 allele means the tag is
present both while the NICD is fused with the transmembrane domain and after NICD cleavage. Thus,
one would expect some fraction of GLP-1::V5 to be detected at the membrane and some in the nucleus
(although Sorensen et al., 2020 reports only detection of their GLP-1::Myc tagged protein in the germ
cell nuclei, and does not comment on gc nuclear signal detection of the V5 tag, though they see it in
embryos). It is not clear to me what “around the nucleus, forming a boundary” means with respect to
GLP-1 biology in lines 21-22 on page 7.

Part of this issue is methodological. Methods page 17:

6 ...To study the distribution of nuclei, the distance between each
7 nucleus and its closest neighbors were analyzed using Imaris distance function. First, using the
8 3D model of the nuclei, the automated analysis calculated the average of three distances for every
9 nucleus. The mean was used to define the distribution of nuclei in the germline. A smaller distance
10 indicates closer packaging of nuclei at the progenitor zone. To measure the distance between
11 each nucleus and GLP-1, the distance from the center of the nucleus to the nearest GLP-1 surface
12 was measured using Imaris.

Looking at Figure 2L and the Methods above, it appears that the authors are measuring the distance
between artificial surfaces created in Imaris software for the GLP-1::V5 antibody and DAPI signals. Having
used Imaris before, I know that when these surfaces are created, they are tunable. Depending on user

choices, surfaces can be more or less generous with connecting patches of signal in a fluorescence channel. Based on Extended Data Figure 2h, it appears that the authors chose to allow the GLP-1::V5 surface to mostly fill the space among germline nuclei, but this choice is neither described nor justified, and does not reflect the biology of that protein; there is also not an image given that allows the reader to compare the Imaris surfaces with the true signal (extended Data Figure 2h is far too small and low-res to make that comparison).

Is this GLP-1::V5 Imaris surface based on signal at the germ cell membrane, in which case the measures of nucleus to GLP-1 signal = distance from nucleus to membrane (a proxy for cell size)? Signal-active GLP-1/NICD would be nuclear; complicating matters is the fact that GLP-1 (including GLP-1::V5) also contains a PEST tag, which targets it for proteolytic degradation, so it is quite likely that the actively signaling GLP-1 proteins do not perdure in the nucleus.

To remedy this issue, I recommend a detailed description and justification of how surfaces were generated, including examples of different surfaces that were rendered with different settings, and why these surfaces were used for measurements instead of primary signal itself. An explanation of GLP-1/Notch signaling and the relationship between Notch protein localization and function should be added. Basically it needs to be clear: what is this measuring and why?

NOTE: The source of *glp-1::V5*, Sorensen et al. 2020, is cited in-line as ref 29, but appears in References as 34. Please double check and update references accordingly.

Concern 3:

Most hits are CLECs? That is what jumps out at me most from Figure 5b. What is the evidence for CLECs being part of the matrisome?

There is no literature cited on work on this gene family in *C. elegans*, for example this recent study by Pees et al., (2021) in *Plos Pathogens* doi: 10.1371/journal.ppat.1009454 which reports that feeding behavior is affected by some *clec* gene RNAi.

Germ cell proliferation is exquisitely sensitive to starvation. If *clec* RNAi is causing feeding defects and therefore triggering a starvation-like state, I would be highly concerned that this is the main relationship between the distal gonad and the *clec* genes. In fact, starvation is a likely cause of non-cell autonomous reduction in distal germline proliferation in general, but there is no discussion of starvation in the manuscript at all.

This statement regarding CLECs on page 11 is not clear to me:

12 ...Although

13 the presence of a large number of CLECs has not been verified experimentally in *C. elegans*, we
14 systematically silenced each *CLEC* gene and analyzed the germline.

To what does “presence” refer? Is there doubt about their presence (that is, their proper annotation) in the genome? EST/transcriptomic evidence that they are really transcribed genes? Their expression in the

germline? General lack of study of the gene class in *C. elegans*? Since these are most of the hits obtained, a more thorough discussion of CLECs is in order.

Concern 4:

I note that laminin is not categorized in Figure 5b as a basement membrane component. Why not? Along with collagen, it is one of two major BM scaffold constituents. Larval laminin RNAi knockdown is known to cause gonad basement membrane rupture and germ cell displacement (Gordon et al., 2018 Current Bio). Did it cause BM rupture in your hands?

Page 12:

7 ...Silencing the matrix structural protein laminin (epi-1, lam-1 and lam-3) caused defects
8 at the distal end and an increase in apoptosis (Extended data fig. 4f).

Literature

In addition to improving the use of the literature in the sections on GLP-1, the zig-zag gonad, CLECs, and laminin (discussed above), I note a general lack of scholarship in the manuscript. First and most importantly, where did these 443 genes come from? Information on choice of candidates belongs in the Introduction (why these are important genes), Results (what the first experiment is), and in great detail in the Methods (how/why chosen based on what references from what database, etc).

A 3-paragraph introduction that never defines the biological roles of ECM or any of its constituents is not acceptable, in my opinion. Discussion of matrix biology in general is mostly absent, and discussion of the basement membrane is entirely absent. The basement membrane is a very important ECM type, and basement membrane is known to entirely surround the *C. elegans* gonad.

This recent paper tagging 29 basement membrane proteins and receptors should be referenced, especially since the gonad was one of the organs specifically investigated in the study. Keeley et al., 2020 (BM toolkit) DOI: 10.1016/j.devcel.2020.05.022

There is no discussion of what is already known of the relationship between ECM and germ cells in *C. elegans*, for example this recent paper: McIntyre and Nance, 2023 Development DOI: 10.1242/dev.201640 "Niche cells regulate primordial germ cell quiescence in response to basement membrane signaling."

Finally, the study design appears to be derived from Green et al. (2011) in Cell, which uses the same germline-marked strain to assess the germline phenotypic consequences of RNAi KD of 554 essential genes. What is the overlap between the matrisome genes and Green's 554 genes? Where overlap exists, do the results agree? Citations of this prior study are scanty.

Odds and ends:

Page 2 line 18: "As the cells move proximally, they enter meiosis in the transition zone"

The transition zone does not mark meiotic entry, but prophase of meiosis 1; meiotic entry begins distal to the TZ with meiotic S-phase.

“germ tube” is an unusual term for *C. elegans*; suggest “gonad”
Proofread/line edits needed (page 9 line 31 for example

Reply to reviewers

We thank the reviewers for their insightful comments. We have addressed all the questions from the reviewers below. In addition, we identified a mistake due to an error in one of the spreadsheets. This has been corrected now. As a result, specific phenotypes for five genes are changed. However, the total number of genes that showed at least one phenotype after RNAi remains the same. We amended the manuscript as required to reflect the changes. We apologize for this unintentional error.

Reply to specific comments

Reviewer #1 (Remarks to the Author):

1). I will comment on Figure 2 because I found it most challenging and because it had issues that reoccurred in other figures. The single purple dot in Figure 2a (mis-labeled as 2b), implies that there was only one control for all experiments. I presume this was not the case. Showing variability, which is likely considerable, could be useful. "Not significant" genes could have been caused by consistent lack of effect or by variable effects. These two categories may be meaningfully different. I suggest this should be displayed, possibly in the supplement, and discussed. I did not understand the point of color-coding. If genes were categorized as "progenitor" or "transition" based on overall phenotypes, wouldn't they preferentially affect numbers of cells in the PZ and TZ, respectively? Dot size and particularly opacity were hard to see, except in the most obvious cases. The Venn diagram looks strange since 39 isn't that much greater than 32. Plotting 2a differently and maybe as separate panels may help.

Reply: Figure 2a and other graphs are now replaced with simplified graphs. Non-significant gene knockdowns are removed from the new graphs. Controls from individual experiments are shown in new supplementary figures 3 & 4, and Supplementary Table 5. This is explained in figure legends (Page 29 lines 3-6) The variable effect of gene knockdowns is now explained in text (Page 6 lines 15-17) and new supplementary figures 3 & 4, and Supplementary Table 5.

2) I did not understand the message of Figure 2b.

Reply: Figure 2b is now better explained in the text and figure legends (Page 6 lines 22-33; Page 29 lines 7-8, Page 30 lines 1-2).

3) Figures 2c-g showcase how complex the phenotypes are. Even among the select 23 genes with defects in the # of PZ cells, there is a complex and variable pattern of effects on the S phase. Extracting these patterns from 2d-g was not easy. Wouldn't it be clearer to list genes putatively affecting different portions of the S phase under relevant images in 2c? The data currently in 2d-g could go into supplement or be presented in a way that would make it easier to see effects of genes on different portions of the S phase.

Reply: The figures are now simplified using a diagram with gene names showing changes in different stages of the S-phase (new figure 2d). The bar graphs showing the quantification are now moved to supplementary figure 5. Additional information is added to new supplementary Table 5 to reflect the data.

4) In Figures 2a, 2k, and 2m it might make sense to show only those genes that caused significant effects. This would focus analysis and discussion on possible biological insights rather than the number of genes tested.

Reply: We agree with the reviewer and the graphs (new figure 2a, 2h, 2j) are simplified by showing only genes that showed significant effect. The rest of the values are now moved to new supplementary table 5.

5) I found Figures 3-5 generally easier to follow, although some elements in them have the same problems as in Figure 2.

Reply: The graphs now simplified by removing the genes that did not have an impact.

Minor:

6) "...cell behavior analysis of >3500 germlines and >7 million germ cells" seems needlessly granular in the Abstract

Reply: This is now removed from the abstract.

7) I did not follow the point of lines 1-10 at the top of p. 5. Do these percentages really matter? I can see utility of briefly stating autonomous vs. non-autonomous, but am concerned whether we really know expression patterns of the putatively "non-autonomous" genes well enough to make a strong claim. I recommend shortening this long paragraph to the most essential statements and being explicit regarding limitations of such analyses.

Reply: The paragraph simplified by removing percentages as suggested by reviewers (Page 5 lines 22-27). The figure legend is better expanded to explain the data (Page 28 lines 13-17).

Reviewer #2 (Remarks to the Author):

1) There are two limitations of the work: The first is that the study only relies on RNAi and does not present validation of phenotypes either with mutations or conditional AID alleles. While there are a few examples that show consistent phenotypes, there were also several that showed different results, leaving one to wonder. While it would be onerous to ask the authors to make such alleles for all of the genes, they could do this for a few of the genes with novel phenotypes or at a minimum they should acknowledge this more robustly and perhaps add a note somewhere in their tables or figures about which genes have known phenotypes or have phenotypes (such as embryonic lethality) that would preclude analysis by other methods. The second caveat with the work, that the authors acknowledge, is the exposure of RNAi for only 16 hours, which may not have been sufficiently long enough to reduce or eliminate function of stable ECM components. Consistent with this, the RNAi with the strongest phenotypes encode signaling molecules and modifying enzymes not core ECM structural proteins. The short time frame may also have prevented a subset of phenotypes from being observed, e.g. a meiotic defect may interfere with chromatin structure, but this may not be visible until later in the germ line. That said, the authors address this concern in the discussion (although this aspect could be lengthened a bit) and it does not detract from an otherwise ambitious study that moves the field forward and will be of broad interest.

Reply: We performed experiments with genetic mutants to confirm the certain phenotypes of RNAi before submission of the manuscript. In addition, we also repeated some RNAi experiments with larger sample size. We added these additional data for reviewers to refer to in Reviewers figure 1. However, we decided not to include these data in the manuscript due to the broad spectrum of matrisome genes that showed an impact on the germ line. This made it less clear why to focus on any specific genes in this study. In addition, a large number of available mutants or early RNAi (L1 onwards) of matrisome genes with phenotype appears to be sterile or lethal. We also generated a new Supplementary Table 2 showing the

curated impact of matrisome molecules in lethality, sterility, brood size and germline defects. This is added to manuscript (page 4 line 32-35, page 5 line 1).

Reply: The possibility of impact of stable ECM components, late defects and the short RNAi is now addressed in the discussion (Page 13 lines 31-35, Page 14 line 1, Page 15 lines 4-6 and 12-14).

2) Several issues that can readily be addressed would improve the manuscript:

- It would be helpful to see a fluorescent or DIC image of an example of each type of defect. This could be shown together in one supplemental image, but would really help the reader understand the breadth of phenotypes examined from the beginning.

Reply: A new Supplementary Figure 2 is added to the reflect specific phenotype and method of analysis.

3) One general comment about data presentation. While I understand that the imaging of the anterior gonad, puts the distal end to the right, in general since the germ cells move distal to proximal, most papers put the distal end to the left and the oocytes to the right. This would flip the images in Fig 1c and d, make the graph in Figure 1e, read i – viii instead of reverse. I think this would help readers compare the results to other published studies. It would also reverse the image in Figure 5 and associated genes to read left to right from distal to proximal (younger to more developed).

Reply: The orientation of the images is now changed to reflect reviewer comments.

4) The gonad tube defects are confusing since the RNAi exposure began in L4 and persisted for only 16 hours. At these times, only the most distal part of the gonad tube is not fully elongated. So what is the nature of the defect in these cases?

Reply: We agree with the reviewer. It is unlikely that gonad morphology is a developmental issue when RNAi is applied at the late L4 stage. Our previous study has addressed this issue (PMID:34795288), where a mutant of the matrix gene showed gonad development (DTC migration) defects, whereas late L4 RNAi failed to produce the same result. We think the defects we found in the current manuscript are likely due to gonad displacement, rather than a developmental phenotype. Nevertheless, we have now addressed this in the manuscript (Page 5 lines 16-20).

5) It is unclear which of the subset of genes from Fig 1E are being shown in Figure 1F.

Reply: Each graph represent a specific subsets of figure 1f.

Figure 1f left bar graph – Shows the percentage of expressed genes having a phenotype at a particular region.

Figure 1f left table – Shows the percentage of all (443 genes) genes with a location specific expression and/or phenotype. A small percentage of the total genes showed expression and phenotype at the same location. This is expanded to the heatmap on the right. Additional explanations added to the figure legend (Page 28 lines 13-17) and figure legend.

6) Timmons and Fire should be referenced for RNAi feeding. Fraser made the first libraries, but Timmons developed the protocol. Please reference accordingly.

Reply: References (number 28-29) added.

7) Page 5 lines 1-2 suggests that multiple expression datasets were examined for germ line expression but only Tzur 2018 is referenced. Were there other datasets that were examined? If not, why was this the one chosen? These expression data presumably include germ line (i.e. germ cells and DTC and sheath cells. Please make this clear in the text.

Reply: We apologize for the confusion. We should have written 'a dataset' instead of 'datasets'. To our knowledge, there are two datasets including Tzur *et al* 2018 (PMID: 30093412) showed the spatial expression of genes in the germ line. However, the website hosting the second data set is corrupted and the authors were not able to revive the website or send the original data file upon our request.

8) I am confused by the yellow bar along the left edge of figure 2H since the control is at the top, what is the bar marking? It would be nice to specifically label the images based on how you describe them in the text, e.g. "completely disorganized". Are one of the images supposed to represent that class?

Reply: We now removed the yellow bar and added more information to the figure (new supplementary figure 5i) to explain the phenotypes.

9) It is curious in Figure 3f that mutant RNAis cluster with ~30% and ~43% blebbing. Is there something unique about these values?

Reply: The percentages were derived from the total number of germ lines analysed and the defective germlines identified. For examples, if we analysed 7 germlines and 3 were defective, we get ~43% as defective. In several cases, we analysed 7 germlines and we obtained 3 defective ones. Hence, these values appear to cluster around ~43%.

10) The nuclear morphology defect in Figure 3D.d2 looks like a defective chromatin, possibly due to DNA fusions (e.g. mre-11 mutants would look like this). The nucleus, however, is otherwise normal (round, etc). Are all of this class of defects associated with aberrant chromatin morphology?

The phenotypic analysis tool used to study oocyte in this manuscript was developed by Green *et al*, 2011. They classified similar oocyte nuclear morphology defects likely to be associated with chromatin defects. Therefore, it is highly likely that the genes we identified generating same types of defects. We commented on this in the manuscript (Page 14 lines 24-27).

11) In Fig 3D, d3, is it possible that blebbing is not actually from the oocytes but rather defective engulfment of apoptotic nuclei in the proximal gonad just above these oocytes?

Reply: As mentioned in the manuscript, we have made an attempt to connect defects in the oocytes to apoptosis. However, we did not manage to conclusively relate oocyte blebbing to a significant change in apoptosis. We re-checked the data to make sure germ cell corpses are not accumulated after gene knockdowns that also showed oocyte blebbing. We only found 2 genes to have accumulation of apoptotic cells that also had oocyte blebbing. Additionally, the presence of nuclei inside the blebbing is a rare occurrence after the knockdown.

12) Figure 5a should have distal at the top and oocytes at the bottom for consistency between figures (.e.g Figure 1f)

Reply: The figure is edited to reflect this.

13) One caveat to the statement that "Our data show that silencing genes that encode extracellular matrix remodelers had the greatest impact on germ cell development." is that

the RNAi might have been most effective in reducing the function of this class of proteins, which they address below. Perhaps instead state “Our data show that short-term RNAi-induced silencing of genes that encode extracellular matrix remodelers ...”

Reply: The text is now edited to reflect this change (Page 15 lines 4-6).

14) When it is stated that 7 germ lines/RNAi were analyzed, can you confirm that these are from 7 different animals or could 2 germ lines have been from the same worm?

Reply: 7 germ lines from 7 worms. The sample size information is now added to all experiments in methods.

Textual and Figure corrections

15) In the text, the authors should note that they have no idea which cells are actually laying down the ECM components they are studying. In some cases, ECM components, e.g. hemicentrin, are made by cells at a distance and are transported to the gonad.

Reply: We agree with the reviewer regarding the expression of matrisome proteins. We have now added additional text to make it clearer (Page 5 lines 31-33).

16) I cannot read the text in Figure S1g.

Reply: We have increased the text size to improve the visibility (Supplementary Figure 1g).

17) Germline should be two words when used as a noun

Reply: The changes have been made to reflect this.

18) references 14-16 are an odd choice to support the statement “as the cells move proximally, they enter meiosis in the transition zone”. I might instead reference the WormBook reviews on germ line development and possibly meiosis. Similarly ref 18 is not the earliest paper to describe differentiation from pachytene into sperm and oocytes.

Reply: We have added the references (number 11) suggested by the reviewer.

19) p. 3 line 22 “The role of extracellular signaling remains unclear”¹⁷. I think you need a clarification here between cell-cell signaling vs ECM-cell signaling since that is, I think, what you are really getting at. There are excellent descriptions of cell-cell signaling between the DTC and the germ cells to control proliferation and communication between sheath cells and germ cells via junctional molecules.

Reply: The sentence is now changed to clarify the context (Page 3 lines 20-23).

20) p. 3 line 27 “RNA-mediated interference” should just be “RNA interference (RNAi)”

Reply: Changes made (Page 4 lines 5-7).

21) Figure S2C “Time spend in each S phase” should read “Time spent...”

Reply: Changes made (New supplementary Figure 5b).

22) p. 5 lines 24-25 should also include reference to Crittenden and Kimble for size of the distal mitotic region.

Reply: New references (number 33) added.

23) p. 8 line 16, technically cells enter leptotene/zygotene before pachytene. might even be

better to say “they enter meiotic prophase I”

Reply: Changes made (Page 9 line 11).

24) p. 8 line 17-18 not quite sure what is meant by: “During meiotic pachytene, germ cell nuclei organize within plasma membrane boundaries through cleavage furrow formation and cytokinesis^{26,35}”. The nuclei in both the mitotic and meiotic region are ensheathed by membrane, and open only to the shared cytoplasmic in the rachis. Cellularization happens not in pachytene but begins at the onset of diplotene and completed as nuclei reach the diakinesis stage.

Reply: Previous reports showed that germ cells become multinucleated if cytokinesis or cleavage furrow formation are affected during early divisions. It can also happen if the stability of the germline structure (rachis) is affected. We have re-written the section to clarify this (Page 9 lines 13-16).

25) IN extended figure 3 how are you defining multinucleated cells vs multinucleated oocytes?

Reply: A new supplementary figure 2 is now added to explain all phenotypes analysed in this manuscript. This figure shows specific features and criteria of analysis associated with each defect. Oocytes are defined based in the size and position (3 independent cells from the spermatheca) within the germline. We did not observe defects strong enough to have a large cell with multiple nucleus and confuse with oocytes. In addition, the appearance of chromosome/nuclei is different in cells in pachytene and oocytes even when there is a defect. A cell/oocyte is counted multinucleated if there are at least two nuclei (marked by mCherry-histone H2B) is present with in a plasma membrane (GFP fusion that binds PI4, 5P₂) boundary.

26) Figure legend extended data 3 “11% of germlines with showed MNCs after control RNA” should read “Eleven percent of control RNAi germ lines showed MNCs”

Reply: The sentence is now changed (Supplementary page 13 lines 3-4).

27) “However, short-term RNAi of annexins (nex-1, -2, -3 and -4) and hemicentrin (him-4) was insufficient to induce MNC, which contrasts a previous report in null mutant animals.” hemicentrin is referenced, but nex-1 – 4 are not and should be.

Reply: New reference (number 50) is added.

28) p. 9, lines 8-9 The C. elegans germline is self-contained tissue that is capable of cross-talk between distal, proximal and oocyte regions^{12,17}. Ref. 17 here is less appropriate but there should be references to the work from the Greenstein laboratory.

Reply: New references (number 51, 52) are now included.

29) The statement: p. 9 lines 9-10 that “We hypothesized that defects observed in distal and pachytene regions would result in unhealthy oocytes.” is based on many papers that show that upstream mitotic zone and meiotic defects lead to oocyte defects. One example is the him-4 paper previously cites, but there are many others. And technically, this is not true crosstalk (line 8) between the regions, it is simply a developmental progression.

Reply: We changed the sentence to reflect reviewer’s comment and additional references are added. See point 28 above (Page 10 line 6-8).

30) What does GO:0015629 term refer to?

Reply: Added the referred function (Actin cytoskeleton) for the GO term. It was mentioned as ‘cytoskeleton’ by mistake.

31) P. 9, Line 29, there is an extra space before Ref 40.

Reply: Changes made.

32) p. 10, lines 2-3. Please briefly mention how you examined apoptotic nuclei.

Reply: A new Supplementary Figure 2 is added to explain the details of the analysis, which is also added to the methods. Briefly, the apoptotic cells were identified by observing the nuclei both by fluorescent intensity and differential interference contrast (DIC) microscopy.

33) The abbreviation for CLECs is introduced on p. 11, but is used extended data 3 legend which is discussed prior. Please spell out C-type lectin in that figure legend as well.

Reply: Changes made to figure (New Supplementary Figure 8).

34) p, 11 line 29 “To study...” should start a new paragraph for ease of reading.

Reply: Changes made (Page 12 line 36).

35) p 12 line 32 should read “germ cell fates” not “fate”

Reply: Changes made (Page 13 line 29).

36) p. 13, line 15-16 “It is likely short time of gene silencing did not allow other mechanisms take into effect.” is not a sentence.

Reply: The sentence is now edited to better convey the message (Page 14 line 32).

37) p. 13, last line. Hyphenate “hedgehog-related”

Reply: Changes made throughout the manuscript.

38) p. 15 line 3 italicize “E. coli”

Reply: Changes made (Page 16 line 26).

39) p. 15 line 5, please define “RNAi plates” DO these have IPTG to induce the dsRNA? What concentrations.

Reply: Yes, we used 1mM IPTG concentration. Method is now amended to reflect the details (Page 16 line 28).

40) p. 15 line 33, Edu should be EdU and in multiple other places in that paragraph.

Reply: Changes made throughout the manuscript.

41) p, 16 line 23 “cytoskeleton” should be “cytoskeletal”

Reply: Changes made (Page 18 line 31).

42) p. 16 line 29. Mistaken reference and wrong format, “Germlines (7 germlines per RNAi) were extruded from the glp-1::v5(q1000) strain after RNAi. This strain expresses GLP-1 with a V5-tag (29).”

Reply: Changes made (Page 19 lines 1-2). Reference (number 44) amended.

43) p. 17 line 25 “condensed nuclei morphology” should be “nuclear”

Reply: Changes made (Page 20 line 11).

44) Extended figure 4 legend, unclear first sentence.

Reply: New supplementary Figure 9. Figure legend changed to clarify the message (Supplementary data Page 16 lines 2-4).

45) Germ tube vs. germtube. Both are used.

Reply: The term germ tube/germtube is now replaced by ‘gonad’ as per recommendation from reviewer 3.

46) Text is extended fig 5a is hard to read even blown up 250% on my screen.

Reply: The text size is now increased (New Supplementary Fig 10).

47) Fig 5 legend “network showing all the genes regulating the germline and have curated interactors.” do you mean “..and that have curated..”

Reply: The interactions are previously curated in STRING database. This is clarified in the figure legends (Page 36 lines 10). We discarded the predictions from Text mining, Co-expressions, Neighborhood, Gene Fusion and Co-occurrence to generate most realistic data.

Reviewer #3 (Remarks to the Author):

1) Concern 1:

The zig-zag morphology of the distal germline was discovered by Seidel et al. (2018) to be a consequence of germ cell crowding—germline growth through proliferation while packed in the gonad tube caused folds to form in the germline. Therefore, the direction of causality inferred in the Results section running through bottom of 6-top of 7 seems backward:

P6 35 ...As matrisome molecules play crucial roles in cytoskeletal organization, we hypothesized that changes in cytoskeletal structure at the distal end of the germline upon matrisome gene knockdown could influence the germ cell cycle.

Page 13 line 7 in the Discussion repeats this mistaken direction of causation: the cell cycle of mitotic germ cells, which likely depend on the structure of the distal germline.

The authors measure proliferation defects, which (according to the Seidel 2018 model) predict less folding of the distal germline. Why conclude instead that the folding of the germline affects germ cell proliferation? These results need better integration with existing knowledge of germline biology.

Reply: According to Seidel et al., the behaviour of germline stem cells and their daughters is influenced by cell position and can only be understood in the context of tissue architecture. One such factor that determines tissue architecture is the actin cytoskeleton. The extracellular matrix is a key player in maintaining actin integrity. The impact of extracellular molecules in the actin dynamics (both biochemical and biomechanical) has been reported in several studies (PMID: 16845676, PMID: 20154082, PMID: 34831087). The biochemical and biomechanical signalling are known to be instrumental in cell proliferation and differentiation in several models (PMID: 34659876, PMID: 30402108, PMID: 25816885). We think that the change in actin morphology after the knockdown observed in this manuscript is independent of germ cell number changes. The rationale behind this is that the cytoskeleton remains intact after silencing some matrisome genes, even though the PZ cell number was significantly reduced (Example: *skpo-2*, *try-3*). We completely agree with the findings by Seidel et al. (2018) that severe overcrowding germ cells can force the germline to adopt further folding to accommodate excess cells. To show this, they used severe overcrowding (germline tumours) mutants. However, it was not clear if losing the cells can cause actin fold changes in the germ line. Our data show that losing cells does not necessarily alter the cytoskeleton.

Seidel et al. suggested the placement of nuclei, proteins (e.g. analysed in detail for GLD-1) and DTC extension along the cytoskeleton and/or rachis. They predict it can determine the cellular functions tracked along the path. The same paper also showed the interior and exterior placement of nuclei along the folds. Taken together, we hypothesised that the shape of the actin would affect the placement of nuclei and proteins, their interactions with each other and interaction with other influencers, thus affecting the cell cycle and cell division. In addition, the

cell cycle is mechanosensitive and actin plays a significant part in this (PMID: 35491306). These factors directed us to pursue the current direction of analysis. As we know, the signalling in many cases is not unidirectional, especially when it comes to the cellular microenvironment. However, we agree to the possibility of actin changes introduced by the severe overcrowding of cells. We included this and other points discussed here in the manuscript (Page 7 lines 23-28, Page 14 lines 15-22).

2) Concern 2:

This framework for assessing GLP-1 localization described on Page 7 is problematic: 25 ...GLP-1 is also localized around nuclei, forming a boundary (Fig. 2j; Extended data fig. 2h)12,34.

GLP-1 is a Notch receptor; it is a transmembrane protein that (when activated by extracellular ligand binding) cleaves the Notch intracellular domain (NICD), which then translocates to the nucleus and acts as a cofactor with transcriptional regulatory Notch signaling effectors. Depending on the method of visualization, different domains of the GLP-1 protein can be detected. According to Sorensen et al. (2020), the position of the V5 tag in the GLP-1::V5 allele means the tag is present both while the NICD is fused with the transmembrane domain and after NICD cleavage. Thus, one would expect some fraction of GLP-1::V5 to be detected at the membrane and some in the nucleus (although Sorensen et al., 2020 reports only detection of their GLP-1::Myc tagged protein in the germ cell nuclei, and does not comment on gc nuclear signal detection of the V5 tag, though they see it in embryos). It is not clear to me what “around the nucleus, forming a boundary” means with respect to GLP-1 biology in lines 21-22 on page 7.

Reply: We apologise for the incorrect wording of this sentence. We have amended the sentence regarding the GLP-1 localization in the germline (Page 8 line 17-20). We have added GLP-1 images showing nuclear localization Supplementary Fig 6. These changes do not affect the values in the graph. We have already included the GLP-1 staining inside the nuclei while quantifying. The distance values were already accounted as zero micrometres if GLP-1 and nuclei colocalized. That means, there is no distance separating the nucleus and GLP-1 surface or GLP-1 is inside the nucleus. We now marked it on the Figure 2i. We have added further notes on Reviewers figure 2, 3 and 4. A new supplementary Figure 11 is added to the manuscript with explanations.

We removed the analysis of GLP-1 nuclei expression due to the following reasons. First, analysing GLP-1 biology was not our intention. Instead, we wanted to show how matrisome impacts protein distribution in the germline by using the expression of a prominent protein at the distal end. Second, while *glp-1::v5 strain (JK5933)* is an excellent tool to analyse a single cell or embryo as shown previously, it was impossible to manually count all NICD localisations (appear as several punctate structures or very lightly spread) in the nuclei of GLP-1 positive cells in several z-planes for 23 genes. Even the highest quality images did not have enough resolution to perform it automatically. Third, we observed a reduction in GLP-1 distribution length after several RNAi knockdowns, which is likely due to reduced expression of GLP-1. As a result, quantifying nuclear localization is not logical as the difference in nuclei localization could be the secondary effect of total reduction in expression. Nevertheless, we performed a workaround to answer the reviewer’s question. We quantified the number of points where GLP-1 is inside or in direct contact with nuclei. While this does not provide the exact number of active NICDs in the nuclei, it gives an indicator of how many GLP-1 surfaces are in contact with nuclei. This is now added to Reviewers Figure 4.

Part of this issue is methodological. Methods page 17:

6 ...To study the distribution of nuclei, the distance between each nucleus and its closest neighbors were analyzed using Imaris distance function. First, using the 3D model of the

nuclei, the automated analysis calculated the average of three distances for every nucleus. The mean was used to define the distribution of nuclei in the germline. A smaller distance indicates closer packaging of nuclei at the progenitor zone. To measure the distance between each nucleus and GLP-1, the distance from the center of the nucleus to the nearest GLP-1 surface was measured using Imaris.

Looking at Figure 2L and the Methods above, it appears that the authors are measuring the distance between artificial surfaces created in Imaris software for the GLP-1::V5 antibody and DAPI signals. Having used Imaris before, I know that when these surfaces are created, they are tunable. Depending on user choices, surfaces can be more or less generous with connecting patches of signal in a fluorescence channel. Based on Extended Data Figure 2h, it appears that the authors chose to allow the GLP-1::V5 surface to mostly fill the space among germline nuclei, but this choice is neither described nor justified, and does not reflect the biology of that protein; there is also not an image given that allows the reader to compare the Imaris surfaces with the true signal (extended Data Figure 2h is far too small and low-res to make that comparison).

Is this GLP-1::V5 Imaris surface based on signal at the germ cell membrane, in which case the measures of nucleus to GLP-1 signal = distance from nucleus to membrane (a proxy for cell size)? Signal-active GLP-1NICD would be nuclear; complicating matters is the fact that GLP-1 (including GLP-1::V5) also contains a PEST tag, which targets it for proteolytic degradation, so it is quite likely that the actively signaling GLP-1 proteins do not perdure in the nucleus.

To remedy this issue, I recommend a detailed description and justification of how surfaces were generated, including examples of different surfaces that were rendered with different settings, and why these surfaces were used for measurements instead of primary signal itself. An explanation of GLP-1/Notch signaling and the relationship between Notch protein localization and function should be added. Basically it needs to be clear: what is this measuring and why?

Reply: We have explained the question reading NICD nuclear localization above. We can indeed see the NICD expression in the nuclei. Additional explanation is added to methods. Briefly, to avoid subjective analysis of cell number, protein distribution, nuclei distribution and distance analysis, we first created a protocol with set parameters for all 3D model creation and quantification for each experiment. These were established using control germline images and saved in Imaris protocols. We recalled the same parameters for the analysis of each germline imaged after a specific gene knockdown so that data does not depend on user. These parameters will not be universal as it depends on the microscopes, objectives, staining quality and magnification. Each experiment will require optimization by using control germlines. For developing spots with nuclei staining, it is relatively easy as it depends on the nuclei size and shape. We added these parameters to methods for a specific image acquisition setting. The development of surface not only affected by acquisition setting, but also by the quality of staining. Image quality is never the same for two independent experiments. Therefore, placing a single value in the manuscript would be wrong. However, we added more information about the analysis to Reviewer Figure 3 and new Supplementary Figure 11.

In order to see GLP-1 in the nuclei, the images need to be visualized by each z-slice. The surface function only allows showing as 3D with all z-slices (which can be zoomed), though it is challenging to see it visually. Nevertheless, we made an attempt to answers reviewer's questions by generating highly magnified images showing surface generated indeed include nuclear localized GLP-1 (Reviewer figures 3).

NOTE: The source of *glp-1::V5*, Sorensen et al. 2020, is cited in-line as ref 29, but appears in References as 34. Please double check and update references accordingly.

Reply: This is corrected now (Reference number 44).

3) Concern 3:

Most hits are CLECs? That is what jumps out at me most from Figure 5b. What is the evidence for CLECs being part of the matrisome?

Reply: CLECs are shown to be secreted into the extracellular matrix and can co-ordinate cell-matrix interactions (PMID: 29565818). The presence of CLECs in the matrisome is experimentally characterized by matrisome project (PMID: 22159717), which served as the reference for developing the *C. elegans* matrisome by Teuscher *et al* (PMID: 33543001).

There is no literature cited on work on this gene family in *C. elegans*, for example this recent study by Pees et al., (2021) in Plos Pathogens doi: 10.1371/journal.ppat.1009454 which reports that feeding behavior is affected by some *clec* gene RNAi. Germ cell proliferation is exquisitely sensitive to starvation. If *clec* RNAi is causing feeding defects and therefore triggering a starvation-like state, I would be highly concerned that this is the main relationship between the distal gonad and the *clec* genes. In fact, starvation is a likely cause of non-cell autonomous reduction in distal germline proliferation in general, but there is no discussion of starvation in the manuscript at all.

Reply: The literature on CLECs in *C. elegans* is very limited. As mentioned in Pees *et al* CLECs have diverse functions which include, but are not limited to immune response, inflammation and metabolism. A recent paper showed that pathogenic exposure can impact germline (PMID: 37956057). CLECs have known functions in pathogenic response in other models. Taken together, CLECs can have a several functions that can impact germline. While exploring all these is not possible in the current manuscript, we quantified the rate of pharyngeal pumping after silencing several *clec* genes that showed defects in the germline. We only found one out of 9 *clec* genes has an effect on pharyngeal pumping within our RNAi timeframe. We included the results in Reviewers Figure 5. To our knowledge, the current manuscript is the only research that comprehensively shows the impact of CLECs on any germline systems. Studying the mechanistic aspect of this large group of proteins with diverse functions is beyond the scope of this research. However, we amended the discussion with more references (number 71, 72) and additional points to discuss the comments made by the reviewer (Page 15 lines 24-25).

This statement regarding CLECs on page 11 is not clear to me:

12 ...Although the presence of a large number of CLECs has not been verified experimentally in *C. elegans*, we systematically silenced each CLEC gene and analyzed the germline.

To what does “presence” refer? Is there doubt about their presence (that is, their proper annotation) in the genome? EST/transcriptomic evidence that they are really transcribed genes? Their expression in the germline? General lack of study of the gene class in *C. elegans*? Since these are most of the hits obtained, a more thorough discussion of CLECs is in order.

Reply: As explained above, the literature on CLECs in *C. elegans* is limited. However, their expression has been shown (PMID: 26580547). Additionally, we showed the spatial gene expression data of C-type lectin in the germ line in Fig 1f, which is fetched from a

transcriptomic database. To avoid the confusion created by the term 'presence', we have now eliminated the sentence.

4) Concern 4:

I note that laminin is not categorized in Figure 5b as a basement membrane component. Why not? Along with collagen, it is one of two major BM scaffold constituents. Larval laminin RNAi knockdown is known to cause gonad basement membrane rupture and germ cell displacement (Gordon et al., 2018 Current Bio). Did it cause BM rupture in your hands?

Reply: Major matrisome BM proteins such as laminins and collagens are shown separately in figure 5. By comparison with metazoan BMs, several proteins in the matrisome list also could be in the basement membrane. It is not realistic to label any protein non-basement membrane proteins as we do not have data to confirm it. To solve the issue, we changed the labelling 'basement membrane proteins' to 'other basement membrane proteins'. This includes the proteins that do not belong to rest of the groups and still characterised a BM component in databases.

There is a crucial difference between the timing of RNAi between this manuscript and Gordon *et al.* Gordon et al performed RNAi from the L1 stage, while the worm is in early stages of larval development. We performed RNAi from late L4 stage. We have not observed germline nuclei dispersion due to membrane rupture as shown by Gordon *et al.*, indicating the BM is still intact in our system.

5) Page 12:

7 ...Silencing the matrix structural protein laminin (epi-1, lam-1 and lam-3) caused defects at the distal end and an increase in apoptosis (Extended data fig. 4f).

Literature

In addition to improving the use of the literature in the sections on GLP-1, the zig-zag gonad, CLECs, and laminin (discussed above), I note a general lack of scholarship in the manuscript. First and most importantly, where did these 443 genes come from? Information on choice of candidates belongs in the Introduction (why these are important genes), Results (what the first experiment is), and in great detail in the Methods (how/why chosen based on what references from what database, etc).

Reply: As mentioned in the original manuscript, the conservation of genes with mammals is the reason for gene selection. We selected 443 conserved genes after excluding nematode specific genes, which will be of broad interest to researchers using different models. We expanded the methods to clarify gene selection (Page 16 lines 8-14).

The first experiment was to create the RNAi library, which is shown in figure 1a. As reviewer 1 pointed out, beyond core genes, the matrisome functions are not well characterised in *C. elegans*. The phenotypic analysis (both distal and proximal) after RNAi were performed without any preselection/priority of gene families (except conservation) or germline regions. The specific phenotypes are listed in supplementary figure 2 were selected for analysis.

Information regarding databases is now added to the methods (Page 17 lines 2-12).

While we strongly agree with the reviewers comments on adding more references and wider information, we were unable to include references for every family of matrisome proteins due to the journal reference and page limit. However, we now included several new key references in the manuscript.

6) A 3-paragraph introduction that never defines the biological roles of ECM or any of its constituents is not acceptable, in my opinion. Discussion of matrix biology in general is

mostly absent, and discussion of the basement membrane is entirely absent. The basement membrane is a very important ECM type, and basement membrane is known to entirely surround the *C. elegans* gonad. This recent paper tagging 29 basement membrane proteins and receptors should be referenced, especially since the gonad was one of the organs specifically investigated in the study. Keeley et al., 2020 (BM toolkit) DOI: 10.1016/j.devcel.2020.05.022. There is no discussion of what is already known of the relationship between ECM and germ cells in *C. elegans*, for example this recent paper: McIntyre and Nance, 2023 Development DOI: 10.1242/dev.201640” Niche cells regulate primordial germ cell quiescence in response to basement membrane signaling.”

Reply: The introduction and discussion are now amended to include these references (number 16, 26). However, we have to keep this to essentials to comply with editorial requirements in word limit. A new Supplementary table 2 is added to discuss the known functions of matrisome in the germline.

7) Finally, the study design appears to be derived from Green et al. (2011) in Cell, which uses the same germline-marked strain to assess the germline phenotypic consequences of RNAi KD of 554 essential genes. What is the overlap between the matrisome genes and Green’s 554 genes? Where overlap exists, do the results agree? Citations of this prior study are scanty.

Reply: While we used a tool by Green *et al.*, our study is different in the following aspects. 1) We analysed the distal germline, which is not part of Green *et al.* 2) Green *et al* did not focus on the matrisome. Only 9 out of 443 genes we studied were overlapping with Green paper. 3) The knockdown method, timing and recovery were very different between the two studies. Green *et al* used RNAi by soaking for 24 hr, followed by 48 hr recovery time whereas this study used 16 hr gene silencing on plates with no recovery time. Therefore, our data might not allow later defects to come into effect and comparison might not be logical. However, we found similarities between two data sets. Green *et al*, showed 8 out of 9 genes has certain germline phenotype. We found five of these genes have at least one germline phenotypes in our study. Similar to Green *et al.*, our data showed apoptosis changes after silencing *zmp-2* and *epi-1*. Green *et al.* showed reduced brood size/partial sterility/decreased oocyte number (*epi-1*, *F29G6.1*, *lam-1*). Our data from the distal germline analysis showed reduced PZ cells, which explains this phenotype. However, we did not some phenotypes as Green *et al*, which is likely due to short RNAi and lack of recovery time. On the other hand, we found additional proximal phenotypes for *mua-3* and *F29G6.1*. We added a new supplementary table 2 with known curated phenotypes (brood size, sterility, lethality and germline defect) for all tested genes, which include the results from Green et al and others.

Odds and ends:

8) Page 2 line 18: “As the cells move proximally, they enter meiosis in the transition zone” The transition zone does not mark meiotic entry, but prophase of meiosis 1; meiotic entry begins distal to the TZ with meiotic S-phase.

Reply: Required changes made in the manuscript. (Page 3 line 18).

9) “germ tube” is an unusual term for *C. elegans*; suggest “gonad” Proofread/line edits needed (page 9 line 31 for example

Reply: Changes made in the manuscript.

REVIEWER COMMENTS

Reviewer #2 (Remarks to the Author):

The revised manuscript and response address most of my initial concerns. There are only minor comments (mostly textual) that should be addressed:

I am not sure I agree with the significance of the analysis on GLP-1 distance presented at the bottom of p 8 – p 9 .

p. 16 line 17, please identify/reference the Ahringer and Vidal RNAi libraries (or other sources) for the RNAi clones used. Including this information in the Suppl. Tables might be useful as well for others interested in using these clones from the available libraries.

Textual:

Figure 1F: It would make much more sense to put the distal region at top and oocytes at the bottom of both heat maps.

p. 5 "...how many matrisome genes we found control the germline are not previously linked with germ cell/gamete development." Should read "...how many of the matrisome genes that we found control the germ line have not been previously linked with germ cell/gamete development.

p. 6 "We focus here on all matrisome genes except C-type LECTins (CLECs),..." Please state the # of matrisome genes rather than "all".

p. 6 "statistically significant differences, large variations in TZ and PZ were observed.." Should read "statistically significant differences, large variations in the number of TZ and PZ nuclei were observed.

p. 6 "We hypothesized that a reason for the changes PZ germ cell number is altered mitotic cell cycle." Should read: "We hypothesized that a possible reason for changes in PZ cell number is altered mitotic cell cycle."

Axes of suppl Fig 5, Fig 2H should be "% germ lines" (plural and 2 words)

Check grammar and writing botm p 8 – top p 9.

p. 9 line 13: "becoming gametes"

p. 14, line 5 "Some germline associated basement membrane constituents are known, which can signal to the distal niche. However, germline associated...". Should be "germline-associated basement membrane constituents that can signal to the distal niche are known. However, germline-associated..."

p. 14, line 8 “long-range”

p. 14, line 22 “elegans PZ, which has not been” should be “PZ that has not been ” (note lack of comma)

p. 19, line 9 “in-built tool” ... do you mean “built-in”?

Reviewer #3 (Remarks to the Author):

I find the background and inclusion of relevant literature to be better in this version. I encourage the journal to allow sufficient space in the word limit for such improvements.

I have two remaining major concerns, the GLP-1 analyses and inference of site of action for the matrisome genes, along with some other notes (below).

Remaining major concern: GLP-1 analyses

Some of the GLP-1 section still seems to me like a dataset in search of a phenotype/question.

I truly appreciate the effort that was made with the beautiful reviewer figures to improve communication about this analysis, though I am still perturbed that they authors really seem to be studying the Imaris surfaces themselves (see this from Methods: “The changes visualized on the generated surface after RNAi were considered as phenotypic differences (Refer to Supplementary Fig. 6 for an example,”) which practice still has the issues that I described previously. What is the biological meaning of the phenotype actually being measured, which is “distance between nuclear surface and GLP-1::V5 antibody stain surface as generated in Imaris with certain parameters”? This is neither a measure of GLP-1 abundance or extent along the axis of the germline (which is shown in Fig 2C and H, and about which I also have questions, below), nor of nuclear packing (the Y axis in Fig 2J), all of which are phenotypes with some biological rationale. Why instead are you measuring this: “the proximity of nuclei to GLP-1 protein (Fig. 2i, 2j).”

Possibilities are that you are seeing GLP-1 protein that is at the membrane where it acts as a receptor, in the nucleus (actively signaling), or else in the cytosol where some fraction is presumably being made and translocated to the membrane, or else is the NICD on its way to the nucleus. Why measure the distance between those populations—lumped together into a single space-filling surface in Imaris—and a germ cell nucleus? And then average them? If this analysis is going to be included, a biologically-informed rationale for carrying it out should be provided.

“...In order to see GLP-1 in the nuclei, the images need to be visualized by each z-slice. The surface function only allows showing as 3D with all z-slices (which can be zoomed), though it is challenging to see it visually. “

Respectfully, this is a backwards way of approaching analysis of fluorescence imaging data. If a feature that is important (nuclear GLP-1 in this case) is not visible when surfaces are rendered in Imaris, then one should not render surfaces in Imaris to analyze the data. These surfaces are not the data, and if they fail to capture the relevant features of the data, they are useless as representations.

Furthermore, in Figure 2H, why does the Control sample have a ~15% “defect in qualitative GLP-1 distribution”, and the *zmp-3* RNAi looks better than the control on that axis? Doesn't that mean your qualitative scoring scheme performs poorly? Additionally, this qualitative scheme is not described in the Methods other than “visual variations in GLP-1 phenotype”. Better description is necessary.

Remaining major concern: Incorrect inferences of site of action

Page 15 lines 14-15

We found that growth factors and their likely receptors (proteoglycans) function in the distal to pachytene region of the germ line, aligning with their previously ascribed roles^{6,7}.

This reveals faulty logic. Just because an effect is seen on a particular germline region does NOT mean that a protein functions in that region. In fact, reference 7 in that sentence (by many of the same authors) made the very important discovery that the proteoglycan syndecan encoded by *sdn-1* functions in the somatic gonadal sheath. It is not reported to function in the distal to pachytene region of the germline.

The same issue is also present at the end of the intro:

“...location-specific roles during the formation of gametes from undifferentiated germ cells.”

The results show the location-specific *effects* of knockdown, not roles or sites of function of the genes. Cell-specific loss of function or rescue can identify where the genes act, but global knockdown phenotypes cannot.

Results section headers are better, as the language there is about how the genes “control germ cell behavior” in different regions of the germline, which is accurate given the experiment performed and its results.

Relatedly, there's an important typo page 5 line 24: expressed in the germ line (including distal tip cell and sheath cells

This should read “expressed in the gonad”, as the DTC and sheath cells are not part of the germline. The rest of this section needs a careful reread to make sure “germline” is not used when “gonad” is intended.

Other points:

RE my question about the matrisome gene selection:

I see from the new methods section on gene selection that prior paper annotated the *C. elegans*

matrisome. The intro now says this:

In silico characterization of matrisome proteins reveals significant conservation between *C. elegans* and other organisms²⁷. Here, we used RNA interference (RNAi) to systematically profile the function of 443 conserved matrisome genes (see methods), identified by in silico analysis, in the *C. elegans* germ line²⁷.

I suggest rephrasing like "...443 genes from the *C. elegans* matrisome²⁷ with orthologs in mammals (see Methods)."

In the methods, I suggest describing this "in silico analysis" precisely and with relevant details. Were these 1:1 orthologs? Conservation at the family level? Using BLAST? Which mammalian genome? These are not trivial questions.

CLECS

Thank you for looking at pharyngeal pumping! If possible, I'd encourage you to include this in the manuscript proper, instead of only the reviewer figures. I also would like to see an explicit statement that systemic effects of RNAi on for example feeding behavior or other physiological changes inducing starvation have the potential to alter the germline, perhaps on page 5 line 29-33 where you discuss non-cell autonomous regulators of the germline.

germline or germ line, but be consistent please

We thank the reviewers for their insightful comments. We edited the manuscript (shown in green) to reflect the comments from reviewers. We addressed each specific comment below and hope that you are now in a position to accept it for publication.

REVIEWER COMMENTS

Reviewer #2 (Remarks to the Author):

1) I am not sure I agree with the significance of the analysis on GLP-1 distance presented at the bottom of p 8 – p 9.

Reply: We now removed the distance analysis between GLP-1 and nuclei from the manuscript to reflect reviewer's comments.

2) p. 16 line 17, please identify/reference the Ahringer and Vidal RNAi libraries (or other sources) for the RNAi clones used. Including this information in the Suppl. Tables might be useful as well for others interested in using these clones from the available libraries.

Reply: We added a new supplementary table 11 to include the clones from available in Ahringer libraries, which we used for matrisome RNAi. Reference is added to the manuscript. Page 16 line 24-25

Textual:

3) Figure 1F: It would make much more sense to put the distal region at top and oocytes at the bottom of both heat maps.

Reply: Figure 1F changed as suggested

4) p. 5 "...how many matrisome genes we found control the germline are not previously linked with germ cell/gamete development." Should read "...how many of the matrisome genes that we found control the germ line have not been previously linked with germ cell/gamete development.

Reply: Text edited to reflect reviewer's comment. Page 5 line 35, Page 6 line 1

5) p. 6 "We focus here on all matrisome genes except C-type LECTins (CLECs),..." Please state the # of matrisome genes rather than "all".

Reply: Text edited to reflect reviewer's comment. Page 6 line 13

6) p. 6 "statistically significant differences, large variations in TZ and PZ were observed.." Should read "statistically significant differences, large variations in the number of TZ and PZ nuclei were observed.

Reply: Text edited to reflect reviewer's comment. Page 6 line 18-19

7) p. 6 "We hypothesized that a reason for the changes PZ germ cell number is altered mitotic cell cycle." Should read: "We hypothesized that a possible reason for changes in PZ cell number is altered mitotic cell cycle."

Reply: Text edited to reflect reviewer's comment. Page 6 line 25

8) Axes of suppl Fig 5, Fig 2H should be "% germ lines" (plural and 2 words)

Reply: Supplementary figure 5j edited to reflect reviewer's comment. Figure 2h is now recreated as supplementary figure 6b-c.

9) Check grammar and writing botm p 8 – top p 9.

Reply: New text added.

10) p. 9 line 13: “becoming gametes”

Reply: Text edited to reflect reviewer’s comment. Page 9 line 11

11) p. 14, line 5 “Some germline associated basement membrane constituents are known, which can signal to the distal niche. However, germline associated...”. Should be “germline-associated basement membrane constituents that can signal to the distal niche are known. However, germline-associated...”

Reply: Text edited to reflect reviewer’s comment. Page 14 line 9-10

12) p. 14, line 8 “long-range”

Reply: Text edited to reflect reviewer’s comment. Page 14 line 12

13) p. 14, line 22 “elegans PZ, which has not been” should be “PZ that has not been ” (note lack of comma)

Reply: Text edited to reflect reviewer’s comment. Page 14 line 26

14) p. 19, line 9 “in-built tool”... do you mean “built-in”?

Reply: We did not create the tool, but it is part of the software.

Reviewer #3 (Remarks to the Author):

Remaining major concern: GLP-1 analyses

Some of the GLP-1 section still seems to me like a dataset in search of a phenotype/question.

15) I truly appreciate the effort that was made with the beautiful reviewer figures to improve communication about this analysis, though I am still perturbed that they authors really seem to be studying the Imaris surfaces themselves (see this from Methods: “The changes visualized on the generated surface after RNAi were considered as phenotypic differences (Refer to Supplementary Fig. 6 for an example,)” which practice still has the issues that I described previously. What is the biological meaning of the phenotype actually being measured, which is “distance between nuclear surface and GLP-1::V5 antibody stain surface as generated in Imaris with certain parameters”? This is neither a measure of GLP-1 abundance or extent along the axis of the germline (which is shown in Fig 2C and H, and about which I also have questions, below), nor of nuclear packing (the Y axis in Fig 2J), all of which are phenotypes with some biological rationale. Why instead are you measuring this: “the proximity of nuclei to GLP-1 protein (Fig. 2i, 2j).”

Possibilities are that you are seeing GLP-1 protein that is at the membrane where it acts as a receptor, in the nucleus (actively signaling), or else in the cytosol where some fraction is presumably being made and translocated to the membrane, or else is the NICD on its way to the nucleus. Why measure the distance between those populations—lumped together into a single space-filling surface in Imaris—and a germ cell nucleus? And then average them? If this analysis is going to be included, a biologically-informed rationale for carrying it out should be provided.

“...In order to see GLP-1 in the nuclei, the images need to be visualized by each z-slice. The surface function only allows showing as 3D with all z-slices (which can be zoomed), though it is challenging to see it visually. “

Respectfully, this is a backwards way of approaching analysis of fluorescence imaging data.

If a feature that is important (nuclear GLP-1 in this case) is not visible when surfaces are rendered in Imapris, then one should not render surfaces in Imapris to analyze the data. These surfaces are not the data, and if they fail to capture the relevant features of the data, they are useless as representations.

Reply: We believe there was a miscommunication. We did not mean to imply that nuclear GLP-1 is not rendered into a 3D model. We simply wanted to inform the reviewer that the rendered images might not be visible to the reviewer just by looking at a 2D image from a PDF.

However, the analysis of the distance between GLP-1 and nuclei appears to create confusion in the manuscript. In addition, Reviewer 2 does not see the need for this analysis in the current manuscript. After considering the comments from both reviewers, we excluded the distance analysis between GLP-1 and nuclei from the manuscript.

16) Furthermore, in Figure 2H, why does the Control sample have a ~15% “defect in qualitative GLP-1 distribution”, and the *zmp-3* RNAi looks better than the control on that axis? Doesn't that mean your qualitative scoring scheme performs poorly? Additionally, this qualitative scheme is not described in the Methods other than “visual variations in GLP-1 phenotype”. Better description is necessary.

Reply: We thank the reviewer for pointing this out. We did observe variations within control germlines. The ~15% is generated from a combined 230 germlines, whereas the sample size for gene knockdowns was small. 230 germlines were combined from several individual runs. We found controls with no defects in some of the runs.

We presented the graph in an attempt to summarize a large dataset. We recognize the error in the way this was presented. Therefore, we replaced the graph in 2H with gene names. The corresponding values from each run are now added to supplementary figure 7. We only consider a knockdown significant only if more than half of the germlines are defective, which is substantially higher than any control.

If the reviewer insists, we are willing to remove the phenotypic analysis. However, we respectfully want to point out that the GLP-1 phenotypic characteristics in space (z-plane) could be as important as the extension along the germline axis (Supplementary fig 7). In some cases, the GLP-1 appears as collapsed or broken staining in the germline. The differences observed in the surfaces are generated due to phenotypic differences in staining. Specific differences are not easy to visually compare from the original staining. In our experience, developing a 3D model from staining with defined parameters improves this and avoids subjective bias. We and others have used similar method to study protein distribution/tissue structure (PMID: 30893592, PMID: 35776645, PMID: 35199030)

Remaining major concern: Incorrect inferences of site of action

17) Page 15 lines 14-15

We found that growth factors and their likely receptors (proteoglycans) function in the distal to pachytene region of the germ line, aligning with their previously ascribed roles^{6,7}.

This reveals faulty logic. Just because an effect is seen on a particular germline region does NOT mean that a protein functions in that region. In fact, reference 7 in that sentence (by many of the same authors) made the very important discovery that the proteoglycan syndecan encoded by *sdn-1* functions in the somatic gonadal sheath. It is not reported to function in the distal to pachytene region of the germline.

Reply: We apologize for the confusion. We did not intend to say they act on a specific location. Instead, we wanted to point out the location of impact of these molecules in the germline. The text is now edited to reflect reviewers' comment. Page 15 line 19-23

18) The same issue is also present at the end of the intro:
"...location-specific roles during the formation of gametes from undifferentiated germ cells."

The results show the location-specific *effects* of knockdown, not roles or sites of function of the genes. Cell-specific loss of function or rescue can identify where the genes act, but global knockdown phenotypes cannot.

Results section headers are better, as the language there is about how the genes "control germ cell behavior" in different regions of the germline, which is accurate given the experiment performed and its results.

Reply: We removed the term 'location-specific', Text edited to reflect reviewer's comment. Page 4 line 12

19) Relatedly, there's an important typo page 5 line 24: expressed in the germ line (including distal tip cell and sheath cells

This should read "expressed in the gonad", as the DTC and sheath cells are not part of the germline. The rest of this section needs a careful reread to make sure "germline" is not used when "gonad" is intended.

Reply: Text edited to reflect reviewer's comment. Page 5 line 25

Other points:

20) RE my question about the matrisome gene selection:

I see from the new methods section on gene selection that prior paper annotated the *C. elegans* matrisome. The intro now says this:

In silico characterization of matrisome proteins reveals significant conservation between *C. elegans* and other organisms²⁷. Here, we used RNA interference (RNAi) to systematically profile the function of 443 conserved matrisome genes (see methods), identified by in silico analysis, in the *C. elegans* germ line²⁷.

I suggest rephrasing like "...443 genes from the *C. elegans* matrisome²⁷ with orthologs in mammals (see Methods)."

In the methods, I suggest describing this "in silico analysis" precisely and with relevant details. Were these 1:1 orthologs? Conservation at the family level? Using BLAST? Which mammalian genome? These are not trivial questions.

Reply: The original gene list was produced by Teuscher et al. They have provided detailed methods for the *in-silico* analysis and for generating a gene list. The gene list was created by comparing the human matrisome gene list from the Matrisome Project. *C. elegans* genome was generated from Greenwald Lab OrthoList website (<http://greenwaldlab.org/ortholist/>). They also provided detailed family classification and conservation in the article. It would be inappropriate to present the same information about in silico analysis in the current manuscript as it has already been published and we have referenced their work. However, we added additional information to the methods to describe how we annotated orthologs and the dataset is added to the source file. Briefly,

1) We discarded any gene that was marked nematode-specific by Teuscher *et al* from the analysis. The remaining genes were selected for RNAi.

2) The preliminary ortholog list was generated by DIOPT Version 9.1 - https://www.flyrnai.org/cgi-bin/DRSC_orthologs.pl. This included all genes that were found by at least one prediction program used in DIOPT- Similar method was used by Teuscher *et al*

3) We applied stricter parameters (moderate to high-ranked genes with a weighted score above 5) to generate an ortholog list for GO-based analysis. This was done to avoid the likelihood of exaggerated predictions from GO analysis.

21) CLECS

Thank you for looking at pharyngeal pumping! If possible, I'd encourage you to include this in the manuscript proper, instead of only the reviewer figures. I also would like to see an explicit statement that systemic effects of RNAi on for example feeding behavior or other physiological changes inducing starvation have the potential to alter the germline, perhaps on page 5 line 29-33 where you discuss non-cell autonomous regulators of the germline.

Reply: The pharyngeal pumping data is now added as Supplementary Fig 9. New text added to the manuscript (page 11, line 22-27; Page 5, line 33-35).

22) germline or germ line, but be consistent please

Reply: 'Germ line' was used as a noun. This change was made in the previous revision as recommended by the Reviewer 2.

REVIEWERS' COMMENTS

Reviewer #2 (Remarks to the Author):

I am not sure where the evidence for this statement comes from (not the ref indicated): “Cell proliferation is determined by the time each nucleus spends in S phase (Supplementary Fig. 6)35”. IN fact, G1 and G2 have a greater effect in most cell types. Is there something different about the mitotic zone cells of the worm gonad that make this different?

An alternative hypothesis for decrease PZ nuclei is premature entry into meiosis (P. 6, line 25)

Are the 18 genes that could not be cloned into the RNAi vectors described anywhere. This is useful information for the community to a) know what was not tested and b) know what is hard to clone.

For the cytoskeleton analyses, please describe how the germ lines were fixed. Is it the same as in the DT analysis?

For analysis of MNCs and oocyte defects, please state how the germ lines were extruded and how the procedure was conducted (e.g. include: on agar pads, glass cover slides, in levamisole or azide or serotonin? What buffer was used? Did you dissect on worm at a time and image each individually or did you dissect a number and image over an extended time period (which could in theory affect a phenotype).

For pharyngeal pumping, was the assay done on plates? On food/ off food. Please be specific.

p. 4. Line 32 should read: “only a limited number of...”

p. 5 both lines 27 and 29 start with “however”

p. 7 line 7:m “calcul ated” should be “calculated”

p. 15 line 21 final word should be “germ line”; line 22 “germline-associated” (add hyphen),

p. 15 line 31 “immune responses”

p. 18 line 2 and line 27, “Fluoroshield”

p. 19 line 7 “extruded germ lines were stained using”.

Reviewer #3 (Remarks to the Author):

Resolved major concern: GLP-1 analyses. I think it was a good choice to remove the GLP-1-to-nucleus distance analysis. The current version of Figure 2 (with Figure S7) is more impactful. Now, the significance of the finding—matrisome gene knockdown affects the crucial receptor of stem-cell fate—is

clear and transparent.

A final suggestion: Please clarify the meaning of the Venn diagram components in the Figure 2H legend. The outer oval is $p < 0.0001$? Is that total GLP-1 signal quantification? What is the additional significance of "Significant difference in GLP-1 distribution" in red? And the ">50%" in green?

The legend of Figure S7 needs a check for the correspondence with the panel letters and a bit more explanation. I suggest clarifying which RNAi is shown in the top panels that illustrate phenotypes, and name those phenotypes according to the categories shown in 2H (distribution vs. total GLP-1?).

Resolved major concern: Incorrect inferences of site of action. This issue has been addressed

Resolved my question about the matrisome gene selection.

CLECS I appreciate the inclusion of the pharyngeal pumping analysis and its rationale.

We thank the reviewers for their comments. We edited the manuscript (shown in red) to reflect the comments from reviewers. We addressed final comments below.

Reviewers Comments

Reviewer #2 (Remarks to the Author):

1. I am not sure where the evidence for this statement comes from (not the ref indicated): “Cell proliferation is determined by the time each nucleus spends in S phase (Supplementary Fig. 6)35”. IN fact, G1 and G2 have a greater effect in most cell types. Is there something different about the mitotic zone cells of the worm gonad that make this different?

Reply: We edited the sentence to reflect the reviewer’s comment and added additional references. We agree that G1/G2 show an effect in most cell types. Unlike somatic cells, germ cells in *C. elegans* have a very short or non-existent G1 phase during mitosis in PZ (PMID: 21558371, 26551561, 31796552). Nevertheless, we attempted to allocate the G1 and G2 stages based on nuclear size, but we were unable to measure this convincingly for the entire PZ. As a result, we find it improper to claim a change in G1/G2 in the manuscript and hence only showed S phase stages. It would make sense to analyse the S phase as it is the longest phase and changes are quantifiable with EdU. Hence, we measured the S-phase to show cells are under active nuclear proliferation (DNA synthesis).

2. An alternative hypothesis for decrease PZ nuclei is premature entry into meiosis (P. 6, line 25).

Reply: We edited the manuscript to reflect reviewer’s comment. Page 6, line 25-26.

3. Are the 18 genes that could not be cloned into the RNAi vectors described anywhere. This is useful information for the community to a) know what was not tested and b) know what is hard to clone.

Reply: A new supplementary data 11 is added to include the genes that we were unable to clone.

4. For the cytoskeleton analyses, please describe how the germ lines were fixed. Is it the same as in the DT analysis?

Reply: In both DT analysis and cytoskeletal analysis, we fixed the germline with 4% neutral formaldehyde. This is now better explained in the manuscript (Page 17, line 34; Page 19, lines 9-10). We did not fix the germlines in methanol as long-term 100% methanol fixation can cause F-actin damage. However, a very short 15 second dip in methanol enhanced staining quality without affecting the cytoskeleton. We have used this method to reproduce published cytoskeletal phenotypes (PMID: 29079422), which was confirmed in controls of each experiment.

5. For analysis of MNCs and oocyte defects, please state how the germ lines were extruded and how the procedure was conducted (e.g. include: on agar pads, glass cover slides, in levamisole or azide or serotonin? What buffer was used? Did you dissect on worm at a time and image each individually or did you dissect a number and image over an extended time period (which could in theory affect a phenotype).

Reply: For MNCs and oocyte defects, germlines were not extruded. As mentioned in the manuscript, these phenotypes were measured in live animals. Worms were sedated in levamisole and placed on an agarose pad prior to imaging. Hence the dissection timing is not relevant. The details of sedation and agarose pad use is now added to methods.

6. For pharyngeal pumping, was the assay done on plates? On food/ off food. Please be specific.

Reply: The method is now expanded to show the worms were analyzed on food in plates (Page 20, lines 31-32).

p. 4. Line 32 should read: “only a limited number of..”

Reply: Manuscript edited to reflect reviewer’s comments (page 4, line 32).

p. 5 both lines 27 and 29 start with “however”

Reply: Manuscript edited to reflect reviewer’s comments (page 5, line 29).

p. 7 line 7: “calculated” should be “calculated”

Reply: Manuscript edited to reflect reviewer’s comments (page 7, line 8).

p. 15 line 21 final word should be “germ line”; line 22 “germlines-associated” (add hyphen),

Reply: Manuscript edited to reflect reviewer’s comments (page 5, lines 22-23).

p. 15 line 31 “immune responses”

Reply: Manuscript edited to reflect reviewer’s comments (page 15, line 32).

p. 18 line 2 and line 27, “Fluoroshield”

Reply: Manuscript edited to reflect reviewer’s comments (page 18, lines 9).

p. 19 line 7 “extruded germ lines were stained using”.

Reply: Manuscript edited to reflect reviewer’s comments (page 18, lines 6, 29).

Reviewer #3 (Remarks to the Author):

1. A final suggestion: Please clarify the meaning of the Venn diagram components in the Figure 2H legend. The outer oval is $p < 0.0001$? Is that total GLP-1 signal quantification? What is the additional significance of “Significant difference in GLP-1 distribution” in red? And the “>50%” in green?

Reply: Figure legends expanded to clarify information (Page 29, lines 9-14).

2. The legend of Figure S7 needs a check for the correspondence with the panel letters and a bit more explanation. I suggest clarifying which RNAi is shown in the top panels that illustrate phenotypes and name those phenotypes according to the categories shown in 2H (distribution vs. total GLP-1?).

Reply: Panel letters corrected. Gene names and categories are added to the figure and figure legends. Supplementary information-Page 15, lines 1-3.